# RIP1 autophosphorylation is promoted by mitochondrial ROS and is essential for RIP3 recruitment into necrosome

Yingying Zhang[1,*], Sheng Sean Su[1,*], Shubo Zhao[1], Zhentao Yang[1], Chuan-Qi Zhong[1], Xin Chen[1], Qixu Cai[1], Zhang-Hua Yang[1], Deli Huang[1], Rui Wu[1] & Jiahuai Han[1]

Necroptosis is a type of programmed cell death with great significance in many pathological processes. Tumour necrosis factor-α(TNF), a proinflammatory cytokine, is a prototypic trigger of necroptosis. It is known that mitochondrial reactive oxygen species (ROS) promote necroptosis, and that kinase activity of receptor interacting protein 1 (RIP1) is required for TNF-induced necroptosis. However, how ROS function and what RIP1 phosphorylates to promote necroptosis are largely unknown. Here we show that three crucial cysteines in RIP1 are required for sensing ROS, and ROS subsequently activates RIP1 autophosphorylation on serine residue 161 (S161). The major function of RIP1 kinase activity in TNF-induced necroptosis is to autophosphorylate S161. This specific phosphorylation then enables RIP1 to recruit RIP3 and form a functional necrosome, a central controller of necroptosis. Since ROS induction is known to require necrosomal RIP3, ROS therefore function in a positive feedback circuit that ensures effective induction of necroptosis.

[1] State Key Laboratory of Cellular Stress Biology, Innovation Center for Cell Signaling Network, School of Life Sciences, Xiamen University, Xiamen, Fujian 361005, China. * These authors contribute equally to this work. Correspondence and requests for materials should be addressed to J.H. (email: jhan@xmu.edu.cn).

Necroptosis is a form of programmed cell death characterized by cellular organelle swelling and cell membrane rupture, which is mediated by the necrotic signalling complex necrosome[1–4]. Substantial evidence has accumulated to show that necroptosis is involved in diseases caused by viral and bacterial infections, as well as sterile injury-induced inflammatory disorders[5]. Tumour necrosis factor (TNF) is a physiologically and pathologically significant cytokine and is widely associated with necroptosis. Upon binding to TNF receptor 1 (TNFR1), TNF stimulates the sequential formation of signalling complexes in necroptosis: complex I and necrosome[5–8]. During the process of necroptosis, RIP3 recruits and phosphorylates mixed lineage kinase domain-like protein (MLKL)[9,10]. Phosphorylated MLKL then undergoes oligomerization and translocates to the plasma membrane to execute cell death[11–14].

Protein phosphorylation plays an essential role in regulating diverse cellular processes including TNF-induced necroptosis. It is well known that RIP1, RIP3 and MLKL, three key components in the necroptotic pathway, are phosphorylated during necroptosis execution. The phosphorylation sites in RIP3 and MLKL and the function of their phosphorylation have been well documented[9,15,16]. It is also clear that RIP1 kinase activity is involved in necroptosis[6,17] and that RIP1 can be autophosphorylated[17,18]. However, the precise pathway leading to RIP1 autophosphorylation and its function in necroptosis are still unclear.

Reactive oxygen species (ROS) have long been considered as a driving force for necroptosis and also participate in apoptosis[19,20]. For example, it has been demonstrated that TNF can induce mitochondrial ROS and ROS enhance necrosome formation[21,22]. Either elimination of ROS by scavengers such as butylated hydroxyanisole (BHA), or inhibition of the electron transport chain by inhibitors such as amytal (also known as amobarbital) can inhibit TNF-induced necroptosis[19,23–27]. In addition, the importance of ROS in inducing necroptosis has also been verified *in vivo* in a model of tuberculosis-infected zebrafish[28]. However, BHA has no effect on TNF plus zVAD and Smac mimetics-induced necroptosis in HT-29 cells, suggesting that ROS are not involved in the necroptosis of HT-29 cells[7]. And a recent study showed that deletion of mitochondria by mitophagy does not compromise necroptosis in SVEC or 3T3-SA cells[29].

In this study we first confirmed that mitochondria are essential for TNF-induced necroptosis in the majority of cell types tested. We then discovered that RIP1 can sense ROS via modification of three crucial cysteine residues and its autophosphorylation on S161 is induced subsequently. This phosphorylation event allows efficient recruitment of RIP3 to RIP1 to form a functional necrosome. In short, our data uncovered RIP1 as the primary target of mitochondrial ROS in necroptosis, and solved a long-standing question of why RIP1 kinase activity is required for necroptosis.

## Results

**ROS target site is at or downstream of RIP1.** Published studies suggested that mitochondrial ROS participate in necroptosis in some but not all kinds of cells[7,8,23,25,30,31]. A recent work utilized Parkin-induced mitophagy of mitochondria lacking membrane potential to deplete mitochondria and observed that mitochondria depletion did not compromise TNF-induced necroptosis in 3T3-SA and SVEC cells[29]. Since depletion of respiration chain by ethidium bromide inhibited TNF-induced necroptosis in L929 cells[26], we tested the effect of Parkin-mediated mitochondrial depletion in L929 cells. Benzyloxycarbonyl-Val-Ala-Aspfluoromethylketone (zVAD) was included in the experiment to exclude apoptosis. Depletion of mitochondria was executed successfully as

indicated by the reduction of TOM20 protein level, oxygen consumption and mito-tracker staining (Supplementary Fig. 1a–c). As shown in Fig. 1a, removal of mitochondria by Parkin-mediated mitophagy in L929 cells compromised TNF-induced necroptosis, supporting the idea of cell context dependence of ROS involvement. The incomplete inhibition of necroptosis by mitochondria depletion might be due to the incomplete removal of mitochondria (Supplementary Fig. 1a–c).

BHA is widely used to assay the involvement of ROS in cell death and amytal is often applied to prove the involvement of mitochondrial respiration in cell death[19]. We utilized these inhibitors to verify that ROS involvement in necroptosis is cell context-dependent (Supplementary Fig. 1d–e). Consistent with previous reports[19,25,30,31], we observed inhibition of necroptosis by BHA and amytal in L929 cells, HeLa cells containing overexpressed RIP3 (HeLa-RIP3), U937, A cells overexpressed with RIP3 (A cell-RIP3), N cells, primary peritoneal macrophages, and MEF, but not HT-29 cells, which correlated with whether TNF induced ROS production in the given cell (Supplementary Fig. 1d–e). We also tested the effect of these inhibitors on L929 cells with or without mitochondria depletion, and found that BHA or amytal blocked cell death to the same survival level in mitochondria-depleted and not depleted L929 cells (Fig. 1b), indicating that BHA- or amytal-mediated inhibition overlaps the inhibition mediated by mitochondria depletion. Although BHA has other effect such as inhibition of lipoxygenases[32], to simplify our experimental procedure, we still used BHA and amytal in the rest of this study to evaluate the effect of mitochondrial ROS in necroptosis of L929 cells.

Since artificially induced dimerization/oligomerization of signalling molecules in necroptotic pathway can cause necroptosis[33–35], we used this method to examine whether the targeting site of ROS is downstream of TNFR1, RIP1, RIP3 or MLKL. The dimerization/oligomerization system used in our experiments was based on anti-estrogen 4-hydroxytamxifen (4-OHT)-induced homo-dimerization of hormone-binding domain's G521R mutant (HBD*) of estrogen receptor[36]. Here, we generated expression vectors of HBD* fused with TNFR1 intracellular domain (tTNFR1), RIP1 without C-terminal death domain (RIP1ΔDD), RIP3 RHIM mutant (RIP3-RHIM^mut) or MLKL N-terminal domain (MLKLΔPD) (Fig. 1c), and expressed these fusion proteins in their corresponding knockout (KO) L929 cell lines. Treatment of 4-OHT can efficiently induce oligomerization of tTNFR1 (Supplementary Fig. 1f), RIP1ΔDD (Supplementary Fig. 1g), RIP3-RHIM^mut (Supplementary Fig. 1h) and MLKLΔPD[12]. As anticipated, 4-OHT induced death of cells expressing HBD*-tTNFR1, and both BHA and amytal inhibited tTNFR1 oligomerization-mediated cell death (Fig. 1d). Similarly, oligomerization of RIP1ΔDD-HBD* caused cell death and both BHA and amytal inhibited it (Fig. 1e). In contrast, the oligomerization of RIP3-RHIM^mut-HBD*- or MLKLΔPD-HBD*-mediated cell death was not affected by BHA or amytal (Fig. 1f,g). These data indicated that the targeting site of mitochondrial ROS should be at or downstream of RIP1 oligomerization and upstream of RIP3 in the necroptotic pathway.

**Three cysteines on RIP1 are targeted by ROS.** ROS often modulate protein functions by modifying cysteines(C) such as forming intramolecular disulfide bond within or intermolecular disulfide bond between the given proteins[37]. To examine the involvement of cysteine modification on RIP1 during necroptosis, we analysed migration of RIP1 in SDS-PAGE under reducing and non-reducing conditions. As shown in Fig. 2a, RIP1 was engaged in a disulfide bond-linked high molecular weight complex/

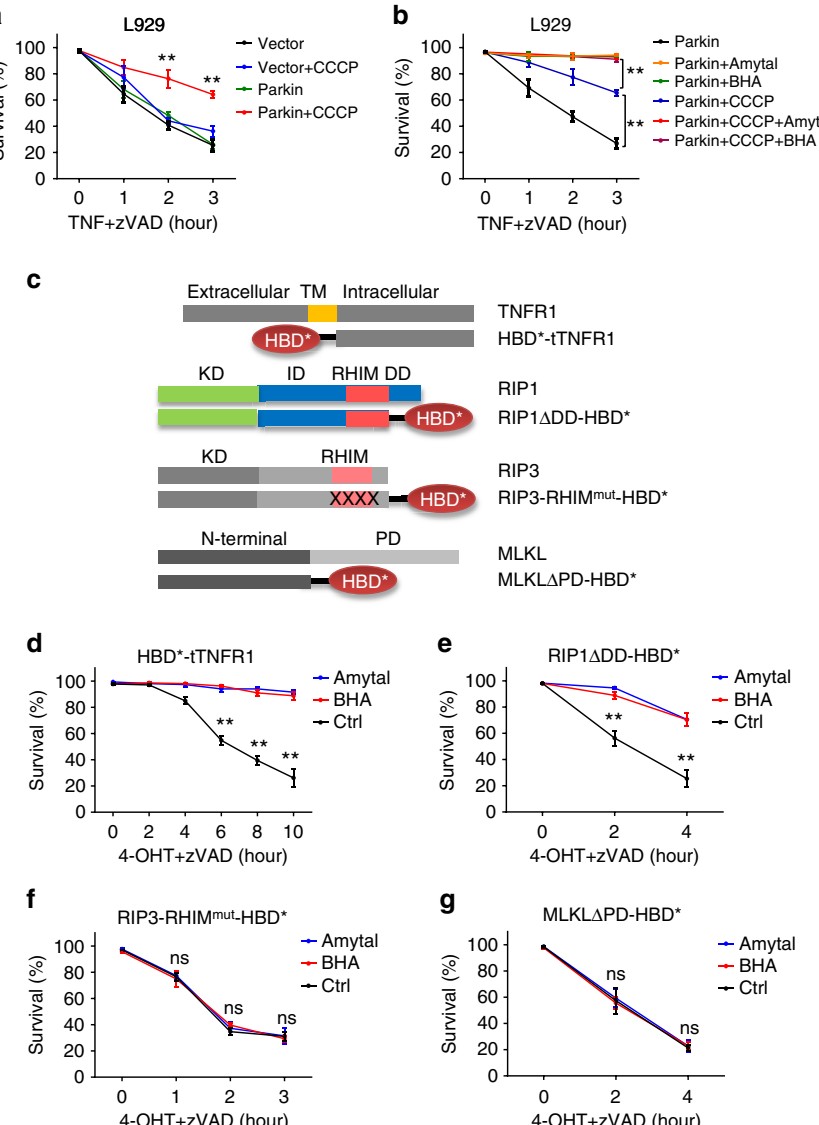

**Figure 1 | Mitochondrial ROS target a site(s) upstream of RIP3 and downstream of RIP1 oligomerization in TNF-induced necroptosis of L929 cells.**
(**a**) Wildtype (WT) L929 cells transfected with Flag-Parkin expression vector or empty vector were treated with CCCP (10 μM) for 48 h. These cells, together with corresponding CCCP non-treated control cells were treated with murine TNF (mTNF) +zVAD for different periods of time as indicated. Survival rate was determined by PI exclusion using flow cytometry. The final concentrations of mTNF and zVAD used in this paper were always 10 ng/ml and 20 μM, respectively, unless specially noted. (**b**) L929 cells expressing Flag-Parkin were treated with CCCP for 48 h. These cells, together with CCCP non-treated control, were incubated with mTNF +zVAD for different periods of time. BHA or amytal was added 1 h before TNF stimulation. Viabilities were measured with PI exclusion. The concentrations of BHA and amytal used in this paper were always 100 μM and 1 mM, respectively. (**c**) Schematic representation of HBD* fused tTNFR1, RIP1ΔDD, RIP3-RHIM$^{mut}$ and MLKLΔPD. HBD* represents the mutated version with G521R, and RHIM$^{mut}$ stands for RIP3 QIG449–451AAA mutation in RHIM domain. TM, KD, ID, DD are short for transmembrane, kinase domain, intermediate domain and death domain, respectively. (**d–g**) Lentivirus encoding Flag-tagged HBD*-tTNFR1, RIP1ΔDD-HBD*, RIP3-RHIM$^{mut}$-HBD* or MLKLΔPD-HBD* was packaged in HEK293T cells, and was used to infect corresponding gene KO L929 cells. These reconstituted L929 cells were then treated with 4-OHT (1 μM) + zVAD for the indicated period of time with or without BHA/amytal. Viabilities were measured with PI exclusion. Data in (**a,b**) and (**d,g**) represented the mean ± s.e.m. of three independent experiments. **$P < 0.01$; ns: no significant difference. See also Supplementary Fig. 1.

aggregate during TNF-induced necroptosis, because this complex cannot be detected in β-mercaptoethanol-treated samples. The formation of complex was inhibited by treating the cells with BHA or amytal. Thus, ROS cause oxidation of RIP1 to form intermolecular disulfide bonds.

To determine which cysteine(s) in RIP1 form(s) disulfide bond, we first mutated all the cysteines on RIP1 to serines, and expressed this All-C-mutant in *RIP1* KO L929 cells to evaluate the role of cysteines in RIP1 in TNF-induced necroptosis. TNF-induced necroptosis in All-C-mutant reconstituted cells was much less than that in Flag-wildtype (WT) RIP1 reconstituted cells and the All-C-mutant cells resisted to BHA and amytal inhibition on necroptosis (Supplementary Fig. 2a), suggesting a ROS-sensing function of cysteines in RIP1. To map the cysteine(s) that respond(s) to ROS, we aligned RIP1 protein sequences from different species and identified C34, C257, C268 and C586 as the conserved cysteines (Supplementary Fig. 2b). Since C34 mutation has been made by others[38], we focused on the

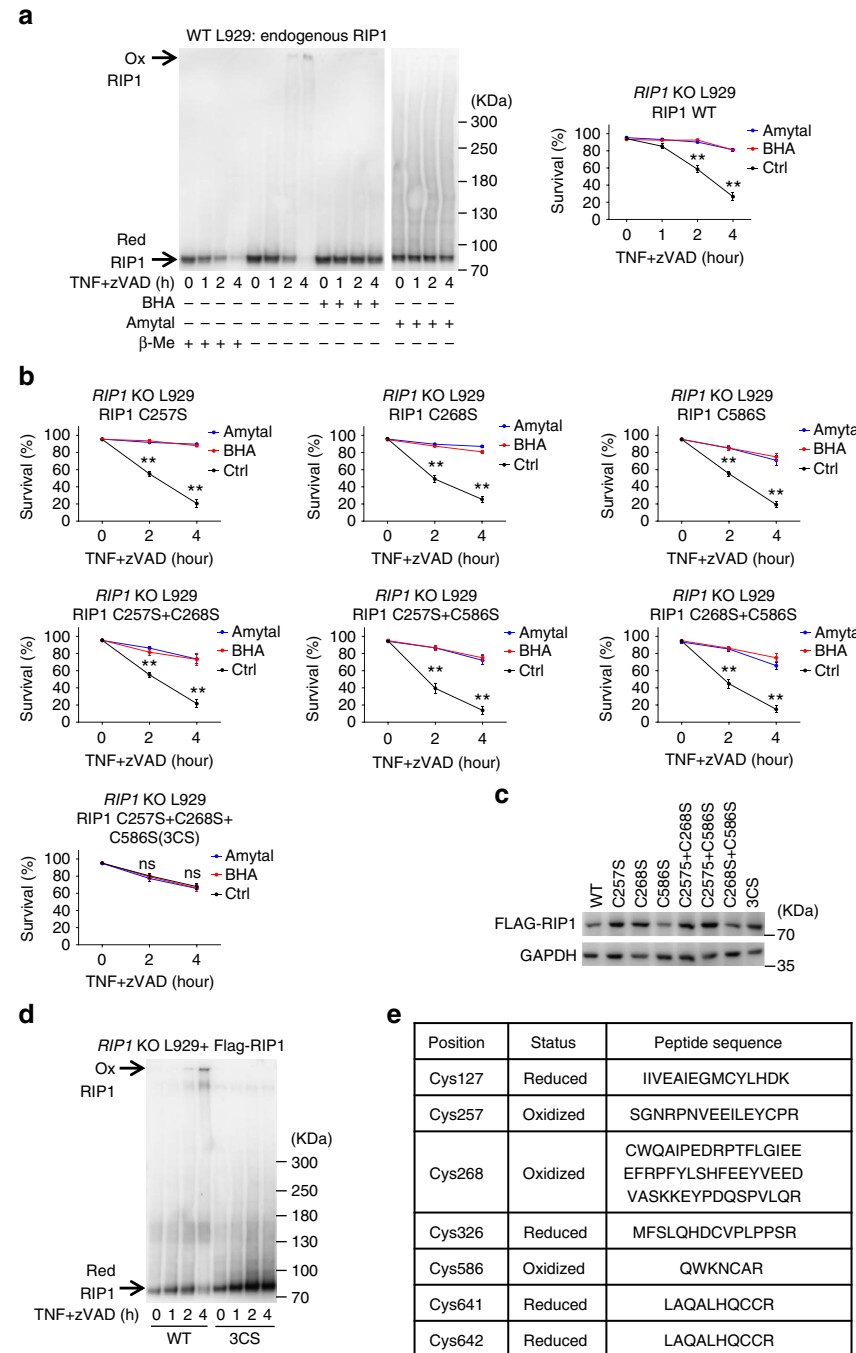

**Figure 2 | C257, C268 and C586 residues in RIP1 are oxidized in response to TNF-induced ROS in L929 cells.** (**a**) L929 cells were treated with mTNF + zVAD for indicated periods of time with or without BHA/amytal. Western blotting of RIP1 with anti-RIP1 antibody was performed under reducing (with β-Mercaptoethanol (β-Me)) and non-reducing (without β-Me) conditions (left panel). Cell viabilities were measured with PI exclusion (right panel). Ox: oxidized form; Red: reduced form. (**b**) The three cysteines, C257, C268 and C586 in RIP1 were mutated to serines individually or in combination as annotated. The Flag-tagged mutants were then expressed in *RIP1* KO L929 cells by lentivirus vector for 24 h. Viabilities were measured by PI exclusion at different time points after treatment of mTNF + zVAD with/without BHA or amytal. (**c**) The expression levels of reconstituted WT and mutant RIP1 in *RIP1* KO cells in (**b**) were measured by western blotting with anti-RIP1 antibody. (**d**) *RIP1* KO cells reconstituted with Flag-RIP1 or Flag-RIP1 3CS as in (**b**) were treated with mTNF + zVAD for indicated periods of time. Western blotting of RIP1 was performed under non-reducing condition. (**e**) Oxidized RIP1 complex was isolated from non-reducing SDS-PAGE and was analysed by MS. The cysteine residues, their redox statuses and the sequences of the tryptic peptides containing those cysteines were summarized. For details see Fig. S2c. Data in (**a**,**b**) represented the mean ± s.e.m. of three independent experiments. **\*\*P < 0.01; ns: no significant difference. Data shown in (**a**,**c**,**d**) are representatives of two to three independent experiments. See also Supplementary Fig. 2.

other three cysteines and generated a series of cysteine-to-serine mutants. Only the simultaneous mutation of C257, C268 and C586 to serines in RIP1 (termed 3CS) makes it insensitive to BHA- and amytal-mediated inhibition of TNF-induced necroptosis (Fig. 2b), though the other mutants were expressed in comparable levels in *RIP1* KO L929 cells (Fig. 2c). Inefficiency of 3CS RIP1 in mediating necroptosis also supports the role of these three cysteines in TNF-induced necroptosis.

To confirm that C257, C268 and C586 are responsible for forming disulfide bonds in TNF-induced high molecular weight RIP1 aggregate, we analysed RIP1 from WT and 3CS RIP1 cells and found the complex was diminished by 3CS mutation (Fig. 2d).

We then analysed the oxidative status of RIP1 cysteines in high molecular weight RIP1 complex and RIP1 monomer. The aggregate and monomer of WT RIP1 were cut from the non-reducing SDS–PAGE gels and subjected to mass spectrometry (MS) analysis. We detected seven cysteine-containing peptides from RIP1 complex and found C257, C268 and C586 were the only three cysteines in their oxidative status (Fig. 2e, Supplementary Fig. 2c). All cysteines detected in the RIP1 monomer were in their reduced status. Collectively, our data demonstrated that C257, C268 and C586 in RIP1 can form intermolecular disulfide bonds, which is required for forming TNF-induced high molecular weight RIP1 complex. This complex/aggregate facilitates necroptosis.

**ROS promotes RIP1 autophosphorylation on S161**. It is known that kinase activity of RIP1 is essential for TNF-induced necroptosis and several phosphorylation sites in RIP1 have been identified[17,18,39,40]. Here, we reconstituted *RIP1* expression in *RIP1* KO L929 cells with WT, D138N (kinase dead mutant caused by a mutation in activation loop), or KK-AT (kinase dead mutant resulted from mutation of K45 and K46 in ATP pocket) *RIP1* expression vector, respectively (Fig. 3a). As anticipated, the kinase inactive forms of RIP1 cannot fully restore RIP1's function as that of WT; however, we could still detect a small amount of cell death upon TNF + zVAD treatment (Fig. 3b). Interestingly, the survival curve of TNF + zVAD-treated kinase-inactive cells was extremely similar to that of WT cells treated with BHA or amytal. Moreover, BHA and amytal had no effect on necroptosis of RIP1 kinase-dead cells (Fig. 3b). These data drop a hint that the function of BHA or amytal is probably to resemble inhibition of RIP1 kinase activity. To test this possibility, we introduced KK-AT mutation in RIP1ΔDD-HBD* and induced oligomerization of the RIP1ΔDD-HBD*-KK-AT in *RIP1* KO L929 cells. The oligomerization of RIP1ΔDD-HBD*-KK-AT caused cell death (Fig. 3c) although it occurred much slower than that induced by the oligomerization of RIP1ΔDD-HBD* (Fig.1e), and BHA and amytal could not influence RIP1ΔDD-HBD*-KK-AT-caused cell death (Fig. 3c). Both RIP1ΔDD-HBD*and RIP1ΔDD-HBD*-KK-AT mediated cell death are necroptosis because they are RIP3- and MLKL-dependent (Supplementary Fig. 3a–c). Thus, ROS's function in necroptosis is very likely to be related to RIP1 kinase activity.

RIP1 kinase activity is required for its autophosphorylation and S14/15, S20, S161 and S166 have been identified as autophosphorylation sites on RIP1[17]. Yet the functional significance of those autophosphorylation sites has been debated since alanine mutations show minimal effect on RIP1-dependent necroptosis[17,18]. S161E mutation does gain resistance to Nec-1 in necroptosis, but it is interpreted as a result of confirmation change rather than the phosphomemetic effect of E to bypass RIP1 kinase activity. Since E or aspartic acid (D) mutation of S or T in RIP1 may allow us to find the phosphorylated RIP1 that can bypass its kinase activity in necroptosis, we collected information from literature, databases and our own MS analyses and found a total of 40S/T as possible phosphorylation sites in mouse RIP1, and mutated them to E. To our surprise, S161E mutant-reconstituted L929 cell line is the only one that can effectively undergo necroptosis and meanwhile is not sensitive to Nec-1 inhibition (Fig. 3d). To confirm that phosphorylation of S161 in RIP1 is inducible in TNF-induced necroptosis, we performed

quantitative MS analyses (Supplementary Fig. 3d,e) and revealed hundreds-fold increase in S161 phosphorylation 2 h after TNF stimulation in L929 cells (Fig. 3e). When BHA or amytal was applied, phosphorylation of S161 was vanished, indicating that ROS promote RIP1 autophosphorylation on S161 (Fig. 3e).

**S161 phosphorylation is essential for RIP1 function**. In reference to the structure of B-RAF, Degterev *et al.* modelled a structure of RIP1 and predicted that S161 would form a hydrogen bond with Glycine (G) 188, which might be important in keeping RIP1 in an inactive status (closed conformation)[17]. Crystal structure of RIP1 is available now, which indicates that S161 locates in the activation loop (T-loop) and its hydroxyl oxygen can form hydrogen bond with carboxyl oxygen of D156[38] (Supplementary Fig. 4a). Accordingly, alanine mutation of S161 should weaken its interaction with D156 and thus increase the flexibility of T-loop (Supplementary Fig. 4b). The flexible structure of S161A mutant should readily be changed to an 'open conformation' and therefore it is still able to mediate TNF-induced necroptosis. Considering that serine possesses structural characteristics such as hydrogen-bond formation and polarity that alanine does not have, other investigators changed the strategy and mutated serine to asparagine (N)[41]. We also used this asparagine mutation to mimic S161 non-phosphorylated RIP1. Asparagine in S161N mutant should be able to form hydrogen bond with carboxyl oxygen of D156 (Supplementary Fig. 4c). Phosphorylation of S161 (or S161E mutation) should destroy the S161-D156 interaction and create an 'open confirmation' of T-loop (Supplementary Fig. 4d). To test this prediction, we compared these mutants with WT and D138N RIP1 for their ability to reconstitute RIP1 function in necroptosis (Fig. 4a,b). Unlike the kinase dead mutation D138N, S161A mutation only slightly reduced RIP1's ability to mediate TNF-induced necroptosis (Fig. 4b), which was consistent with previous reports[17,18]. In agreement with our reasoning, S161N-reconstituted cells were as resistant to TNF-induced necroptosis as D138N-reconstituted cells (Fig. 4b). Furthermore, TNF + zVAD-induced necroptosis in S161E reconstituted L929 cells was slightly quicker than that in WT-reconstituted cells (Fig. 4b). These data suggest that S161 is a functional autophosphorylation site of RIP1, which is required for RIP1 to efficiently mediate necroptosis.

To further demonstrate that the primary function of RIP1 kinase activity in TNF-induced necroptosis is to phosphorylate S161, we introduced S161E mutation in both KK-AT and D138N mutants (KK-AT-S161E and D138N-S161E) and expressed them, respectively, in *RIP1* KO cells (Fig. 4c). In contrast to KK-AT and D138N mutants, which cannot effectively mediate necroptosis, TNF induced quick necroptosis in KK-AT-S161E or D138N-S161E reconstituted *RIP1* KO cells (Fig. 4d). By now, it is clear that S161E mutation in RIP1 can bypass the defect caused by kinase-dead mutation, and the principal role of RIP1 kinase activity in necroptosis is to autophosphorylate S161.

Then we proceeded to examine the effect of BHA and amytal on TNF-induced necroptosis in S161E, S161N, S161A or KK-AT-S161E reconstituted *RIP1* KO cells. We found that BHA and amytal can no longer inhibit cell death when autophosphorylation of S161 in RIP1 was prevented by E or N mutation in both TNF- and RIP1 oligomerization-induced cell death (Fig. 4e–g,i,j), indicating that ROS function to enhance RIP1 kinase activity and the subsequent S161 autophosphorylation. However, BHA and amytal still had inhibitory effect on TNF + zVAD-induced cell death in S161A reconstituted cells (Fig. 4h). This might be due to that alanine mutation makes the T-loop more flexible and allows

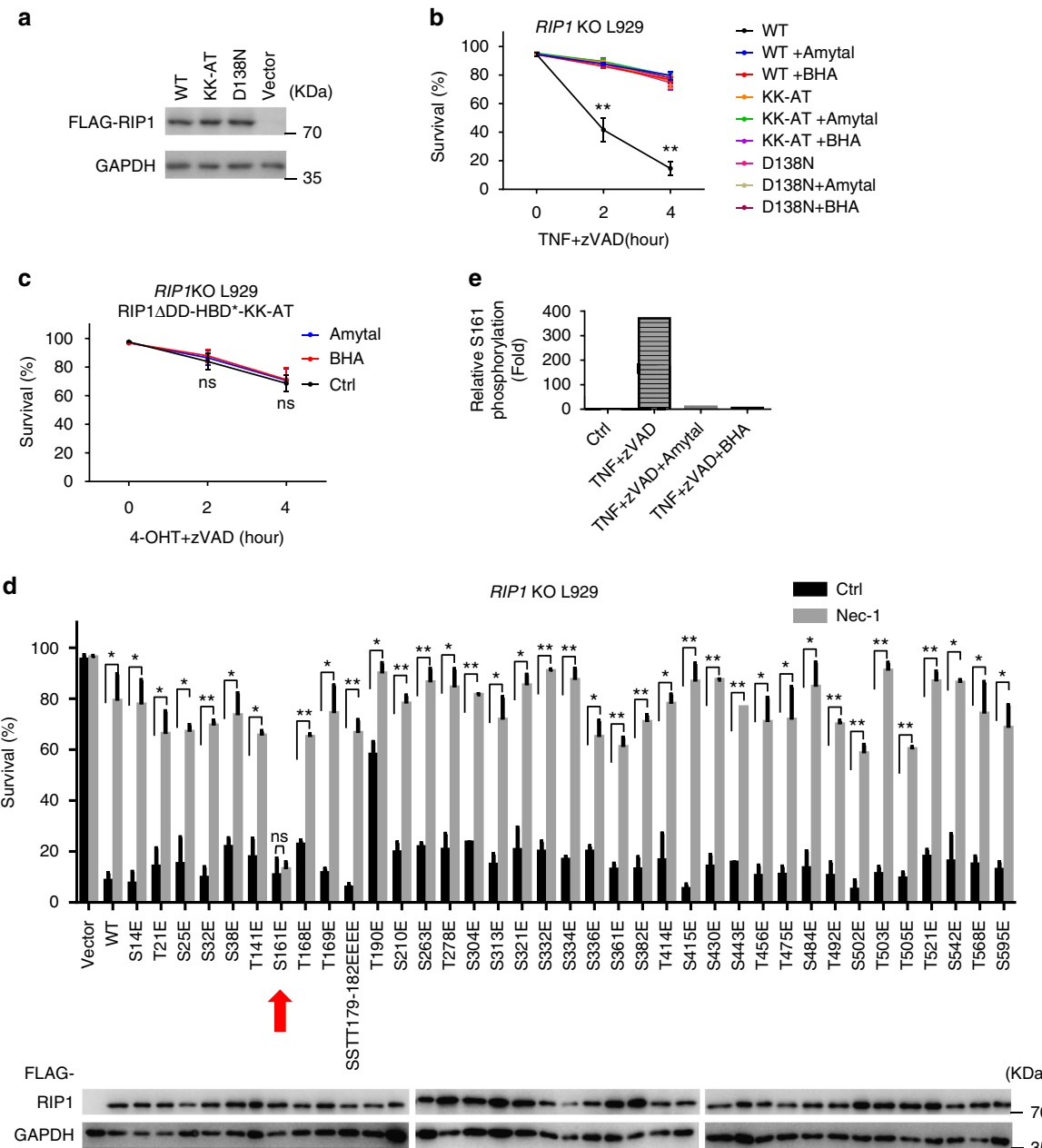

**Figure 3 | Identification of S161 as an autophosphorylation site of RIP1 in response to ROS activation. (a)** *RIP1* KO L929 cells were infected with lentivirus encoding Flag-tagged WT, KK-AT or D138N RIP1, respectively, for 24 h, and then subjected to western blotting for examination of expression level of these proteins. Anti-RIP1 and anti-GAPDH antibodies were used. **(b)** *RIP1* KO L929 cells carrying WT, KK-AT or D138N RIP1 were treated with mTNF + zVAD for time as indicated with/without the presence of BHA/amytal. Viabilities were measured by PI exclusion. **(c)** *RIP1* KO L929 cells reconstituted with Flag-tagged RIP1ΔDD-HBD*-KK-AT were treated with 4-OHT + zVAD for indicated periods of time with or without BHA/amytal. Viabilities were measured by PI exclusion. **(d)** *RIP1* KO L929 cells were reconstituted with RIP1 carrying different serine/threonine mutations and stimulated with mTNF + zVAD for 4 h. Necrostatin-1 (Nec-1) was used at 30 μM to pretreat the cells for 1 h and then kept in the media till viabilities were measured by PI exclusion. RIP1 expression levels were determined by western blotting with anti-RIP1 antibody. **(e)** Flag-tagged RIP1 reconstituted *RIP1* KO L929 cells were treated with mTNF + zVAD for 2 h with or without BHA/amytal followed by Flag immunoprecipitation and targeted MS analysis. Phosphopeptides containing S161 phosphorylation were detected and the MS2 intensities of the S161 phosphopeptide in each sample were extracted and the relative folds were calculated and shown. Data in **(b–d)** represented the mean ± s.e.m. of three and two independent experiments, respectively. *$P < 0.05$; **$P < 0.01$; ns: no significant difference. See also Supplementary Fig. 3.

phosphorylation on other site(s) to partially compensate the function of S161 phosphorylation. Collectively, these data led to a conclusion that ROS enhance S161 autophosphorylation of RIP1 and the S161 phosphorylation is required for RIP1 to efficiently transduce necroptotic signal.

**Cys-mediated RIP1 aggregation enhances S161 phosphorylation.** Since ROS cause RIP1 to form disulfide bond-linked aggregates and ROS enhance S161 autophosphorylation of RIP1, we analysed the effect of disruption of disulfide bond-linked RIP1 aggregate by 3CS mutation on S161 autophosphorylation. *RIP1*

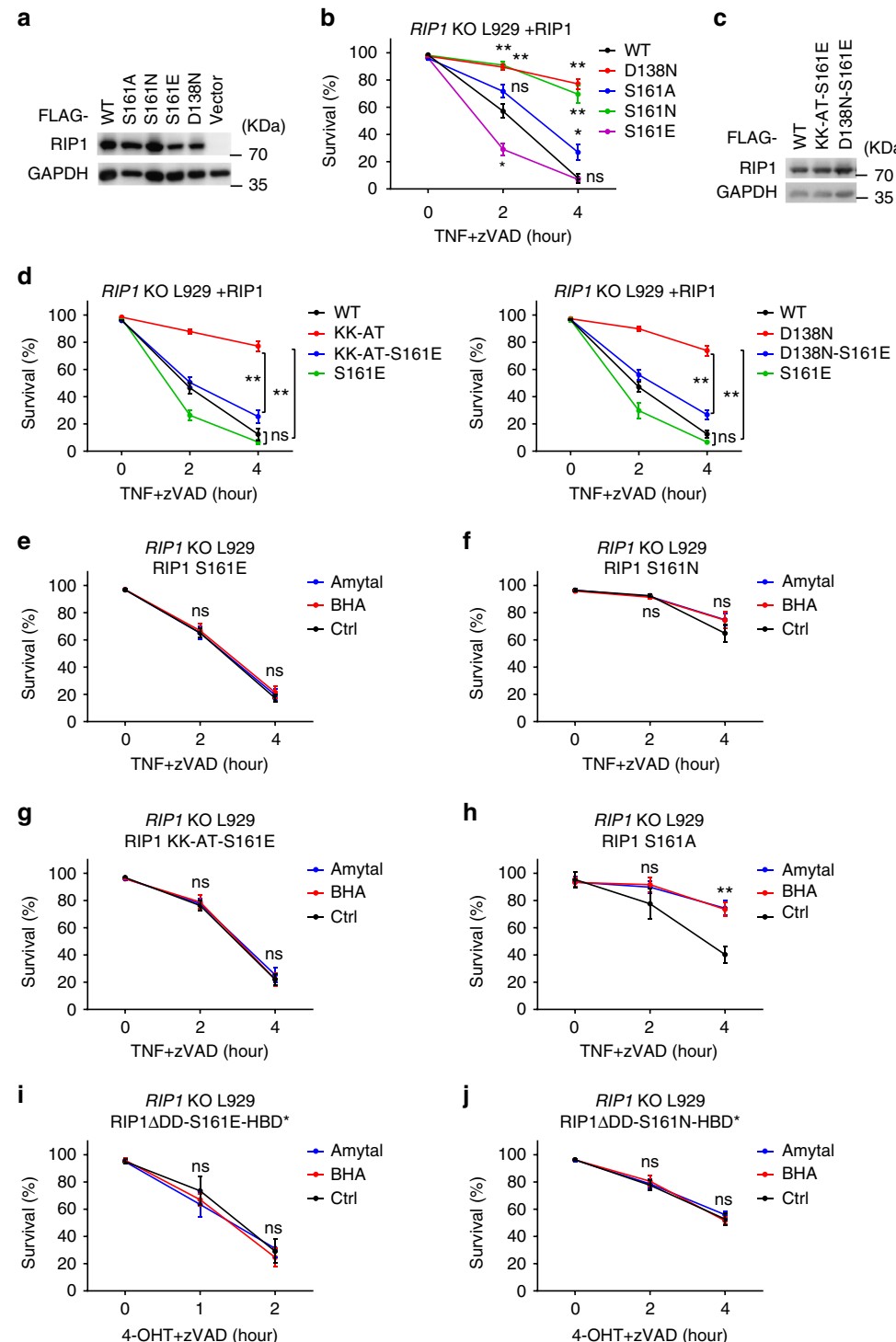

**Figure 4 | ROS promote RIP1 autophosphorylation on S161 and this phosphorylation is essential for RIP1 to effectively transduce necroptotic signal.**
(**a,c**) *RIP1* KO L929 cells were infected with lentivirus encoding Flag-tagged RIP1 WT, D138N, S161A, S161N, S161E, KK-AT-S161E and D138N-S161E, respectively, for 24 h. Expression level of RIP1 was measured by western blotting with anti-RIP1 antibody. (**b,d**) Viabilities of each reconstituted cell line were measured by PI exclusion after mTNF + zVAD treatment for different periods of time. (**e–j**) *RIP1* KO L929 cells were infected with lentivirus encoding Flag-tagged RIP1 S161E, S161N, KK-AT-S161E, S161A, RIP1ΔDD-S161E-HBD* or RIP1ΔDD-S161N-HBD*, respectively, for 24 h. Viabilities of each cell line were determined at different time points under stimulation of mTNF/4-OHT + zVAD with or without BHA/amytal. Data in (**b,d–j**) represented the mean ± s.e.m. of three independent experiments. *$P < 0.05$; **$P < 0.01$; ns: no significant difference. See also Supplementary Fig. 4.

KO L929 cells were reconstituted with WT or 3CS RIP1 expression and the phosphorylation level of S161 was measured using quantitative MS analysis. We found that TNF-induced increase of S161 phosphorylation was blocked by 3CS mutation (Fig. 5a, Supplementary Fig. 5a), indicating that S161 phosphorylation is a downstream event of disulfide bond-mediated RIP1 aggregation.

We then determined whether phosphomimetic S161 mutation, S161E can bypass the inhibitory effect of 3CS mutation on necroptosis. The same as the data shown in Fig. 2a,b, TNF +

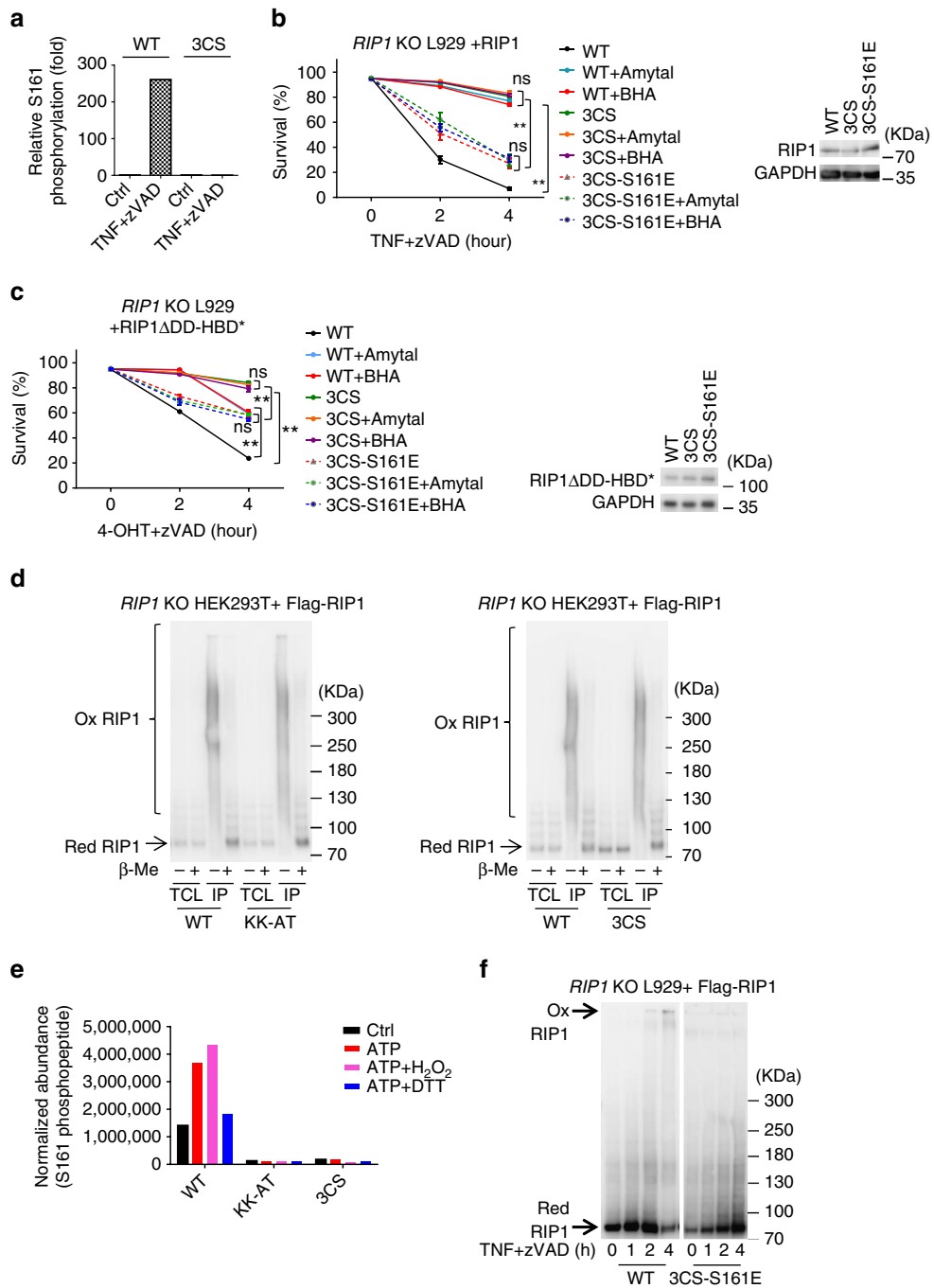

**Figure 5 | 3CS mutation blocks S161 phosphorylation and S161E mutation can bypass the defect of 3CS mutation in necroptosis.** (**a**) Flag-tagged RIP1 WT or 3CS reconstituted RIP1 KO L929 cells were treated with mTNF + zVAD for 2 h followed by Flag immunoprecipitation and targeted MS analysis. Phosphopeptides containing phosphorylated S161 were detected and the MS2 intensities of the S161 phosphopeptide in each sample were extracted and the relative folds were calculated and shown. (**b**) *RIP1* KO L929 cells were infected with lentivirus encoding Flag-tagged WT, 3CS or 3CS-S161E RIP1 for 24 h. Viabilities were determined at different time points after mTNF + zVAD treatment with or without BHA/amytal. The RIP1 expression were determined by western blotting with anti-RIP1 antibody. (**c**) The same as in (**b**) except that the cells were infected with lentivirus encoding Flag-tagged RIP1ΔDD-HBD*, RIP1ΔDD-HBD*-3CS or RIP1ΔDD -HBD*-3CS-S161E for 24 h. Viabilities were determined at different time points after 4-OHT +zVAD treatment with or without BHA/amytal. (**d**) Flag-tagged WT, KK-AT or 3CS RIP1 was expressed in *RIP1* KO HEK293T cells and purified by M2 beads. These proteins were analysed by western blotting with anti-RIP1 antibody under reducing and non-reducing conditions. (**e**) Proteins described in (**d**) were pre-treated with nothing, 10 μM H$_2$O$_2$, or 1 mM DTT for 10 min at 20 °C, and then *in vitro* kinase reactions of these proteins were performed by adding ATP to final concentration of 10 μM and incubated at 30 °C for 30 min. The reaction was stopped by adding TCA to 20%. The TCA precipitates were subjected to targeted MS analysis of S161 phosphorylation. The normalized abundances of S161 phosphopeptide were calculated by normalizing MS2 intensities of the S161 phosphopeptide with that of RIP1 tryptic peptide DLKPENILVDRDFHIK. (**f**) *RIP1* KO L929 cells reconstituted with Flag-RIP1 or Flag-RIP1 3CS-S161E were treated with mTNF + zVAD for indicated periods of time. Western blotting with anti-RIP1 antibody was performed under non-reducing condition. Data in (**b**,**c**) represented the mean ± s.e.m. of two independent experiments. **$P$ < 0.01; ns: no significant difference. Data shown in (**d**,**f**) are representatives of two to three independent experiments. See also Supplementary Fig. 5.

zVAD-induced necroptosis was much less in *RIP1* KO cells expressing 3CS RIP1 compared with the cells expressing WT RIP1 (Fig. 5b). We observed dramatic enhancement of TNF + zVAD-induced necroptosis when S161E mutation was added to 3CS RIP1 (Fig. 5b). As anticipated, BHA and amytal had inhibitory effect on necroptosis in WT but not 3CS or 3CS-S161E RIP1 expressing cells (Fig. 5b). The same result was obtained in RIP1 oligomerization-mediated necroptosis in L929 cells (Fig. 5c). The effect of ROS on RIP1 and the function of RIP1 S161 autophosphorylation should also occur in some other cell lines as similar effects of 3CS, KK-AT, S161E and KK-AT-S161E mutations on necroptosis were observed in MEF cells (Supplementary Fig. 5b).

To examine whether the oxidation of cysteines in RIP1 can enhance S161 phoshorylation *in vitro*, we expressed Flag-tagged WT, KK-AT and 3CS RIP1 in HEK293T cells and purified these proteins, respectively, by anti-Flag M2 agarose beads for *in vitro* kinase assay. The enrichment of RIP1 and its mutant proteins on the M2 beads may have drawn close the protein molecules, therefore the immunopurified proteins formed disulfide bond-linked complexes during the immunopurification (Fig. 5d). The disulfide bonds in those purified proteins were not restricted to those formed by C257, C268 and C586 since the disulfide bond-dependent shifts were also observed in 3CS mutant (Fig. 5d). To eliminate disulfide bonds in the purified proteins, we pre-treated the proteins with Dithiothreitol (DTT). We also included $H_2O_2$ treatment to increase the oxidation of the proteins. The pre-treated and not pre-treated WT, KK-AT and 3CS RIP1 proteins were then incubated with ATP to carry out autophosphorylation reaction, respectively. Quantitative MS was used to determine the levels of S161 phosphorylation after the *in vitro* kinase reaction. The S161 in WT RIP1 complexes was already phosphorylated to a certain level in the ATP-null control sample (Fig. 5e, Supplementary Fig. 5c). This was not surprising because disulfide bonds in the WT RIP1 complexes should contain those formed by C257, C268 and C586, which can promote S161 autophosphorylation (Fig. 5a). *In vitro* incubation of the WT RIP1 complexes with ATP increased S161 phosphorylation (Fig. 5e, Supplementary Fig. 5c). $H_2O_2$ treatment further enhanced S161 phosphorylation in WT RIP1, and in contrast, DTT treatment abolished the *in vitro* kinase reaction-caused increase of S161 phosphorylation (Fig. 5e, Supplementary Fig. 5c). S161 phosphorylation was not detected in control sample of KK-AT mutant and *in vitro* kinase reaction did not lead to S161 phosphorylation, demonstrating that S161 phosphorylation is an autophosphorylation event (Fig. 5e, Supplementary Fig. 5c). S161 phosphorylation was not detected in control sample of 3CS mutant and *in vitro* kinase reaction did not lead to S161 phosphorylation in 3CS complexes, supporting the role of oxidation of three specific cysteines in promoting S161 phosphorylation (Fig. 5e, Supplementary Fig. 5c). Collectively, the *in vitro* kinase assay supports the conclusion that the formation of specific disulfide bond-linked complex promotes RIP1 S161 autophosphorylation.

As 3CS mutant cannot form high molecular weight aggregate (Fig. 2d), but 3CS-S161E did efficiently mediate necroptosis (Fig. 5b,c), we checked whether 3CS-S161E formed aggregate during necroptosis induction. As shown in Fig. 5f, 3CS-S161E cannot form disulfide bond-linked high molecular weight complex. Thus, formation of disulfide bond-linked RIP1 aggregate is to enhance RIP1 S161 phosphorylation and is not required for the signal transduction once S161 is already phosphorylated.

**RIP1 S161 phosphorylation enhances necrosome formation.**
RIP1 participates in the formation of TNFR1 complex (complex I)

and necrosome during necroptosis. We examined TNF-induced complex I formation in *RIP1* KO cells reconstituted with WT, 3CS, 3CS-161E, KK-AT or KK-AT-S161E RIP1 expression, and found no difference among these cell lines in recruiting RIP1, TRAF2 and TRADD to TNFR1 (Supplementary Fig. 6a–b), indicating that RIP1 oxidation and S161 phosphorylation has no effect on complex I formation.

The core components of necrosome are RIP1 and RIP3. To detect necrosome formation more easily, we knocked in a Flag tag at the N-terminal of *Rip1* in L929 cell line by homologous recombination. This Flag-RIP1 L929 cell line behaved similarly to the parental wild-type L929 in TNF-induced necroptosis (Supplementary Fig. 6c). We performed immunoprecipitation with anti-FLAG M2 beads from cell lysate of Flag-RIP1 L929 cells treated with TNF + zVAD for different periods of time, and detected necrosome formation as indicated by co-immunoprecipitation of RIP3 and FADD (Fig. 6a,b). Both BHA and amytal inhibited TNF-induced formation of necrosome, which is consistent with a recent report that BHA inhibited TNF + Smac mimetic-induced necrosome formation[22]. In addition, BHA and amytal had the same effect on TNF alone-induced necroptosis in L929 cells (Supplementary Fig. 6d–e). We also analysed TNF-induced necrosome formation in HeLa-RIP3 cell line, and as predicted, both BHA and amytal had inhibitory effect on human TNF-induced necrosome formation (Supplementary Fig. 6f–g).

Next, we evaluated the effect of various RIP1 mutations on necrosome formation. We immunoprecipitated necrosome from WT, KK-AT or S161N RIP1 reconstituted *RIP1* KO L929 cells (Fig. 6c), and from WT or 3CS RIP1 reconstituted *RIP1* KO cells (Fig. 6d). Analysis of the necrosome revealed that both deprivation of S161 phosphorylation and prevention of disulfide bond formation inhibited necrosome formation (Fig. 6c,d). Interestingly, when phosphomimetic mutation S161E was added to KK-AT or 3CS mutant, the inhibitory effect was bypassed as TNF + zVAD-induced necrosome formation in these cells was restored to a level similar to that of WT RIP1-expressing cells (Fig. 6e,f). Consistent with the notion that S161 phosphorylation is a downstream event of ROS-mediated enhancing effect in the necroptosis pathway, BHA had no inhibitory effect on TNF + zVAD-induced necrosome formation in 3CS-S161E and KK-AT-S161E RIP1 expressing cells (Fig. 6g,h). Collectively, these data demonstrated that S161 phosphorylation of RIP1 positively regulates necrosome formation.

**Rip1 S161 phosphorylation facilitates RIP1-RIP3 co-localization.**
To observe RIP1–RIP3 interaction visually, we used confocal microscopy. To stain RIP1 and RIP3, we generated *RIP1-RIP3*-double-knockout L929 (DKO L929) cell line and reconstituted it with Flag-tagged RIP3 and HA-tagged RIP1. As shown in Fig. 7a, TNF + zVAD induced RIP1 and RIP3 co-localization and both BHA and amytal inhibited RIP1 and RIP3 co-localization. Similarly, no co-localization of RIP1 and RIP3 was observed in cells bearing 3CS (Fig. 7b), KK-AT or S161N mutant (Fig. 7c). In agreement with the co-immunoprecipitation data (Fig. 6), 3CS-S161E and KK-AT-S161E mutant of RIP1 bypassed the kinase-dead defect as we observed co-localization of RIP1 with RIP3 (Fig. 7b,c). Therefore, eliminating RIP1 autophosphorylation inhibits TNF + zVAD-induced co-localization of RIP1 and RIP3.

It is known that necroptosis signalling is mediated by sequential events of RIP1–RIP1 homo-interaction, RIP1–RIP3 hetero-interaction and RIP3–RIP3 homo-interaction[35]. To test whether the lack of RIP1–RIP3 interaction/co-localization observed in Figs 6 and 7 is due to that S161 non-phosphorylated

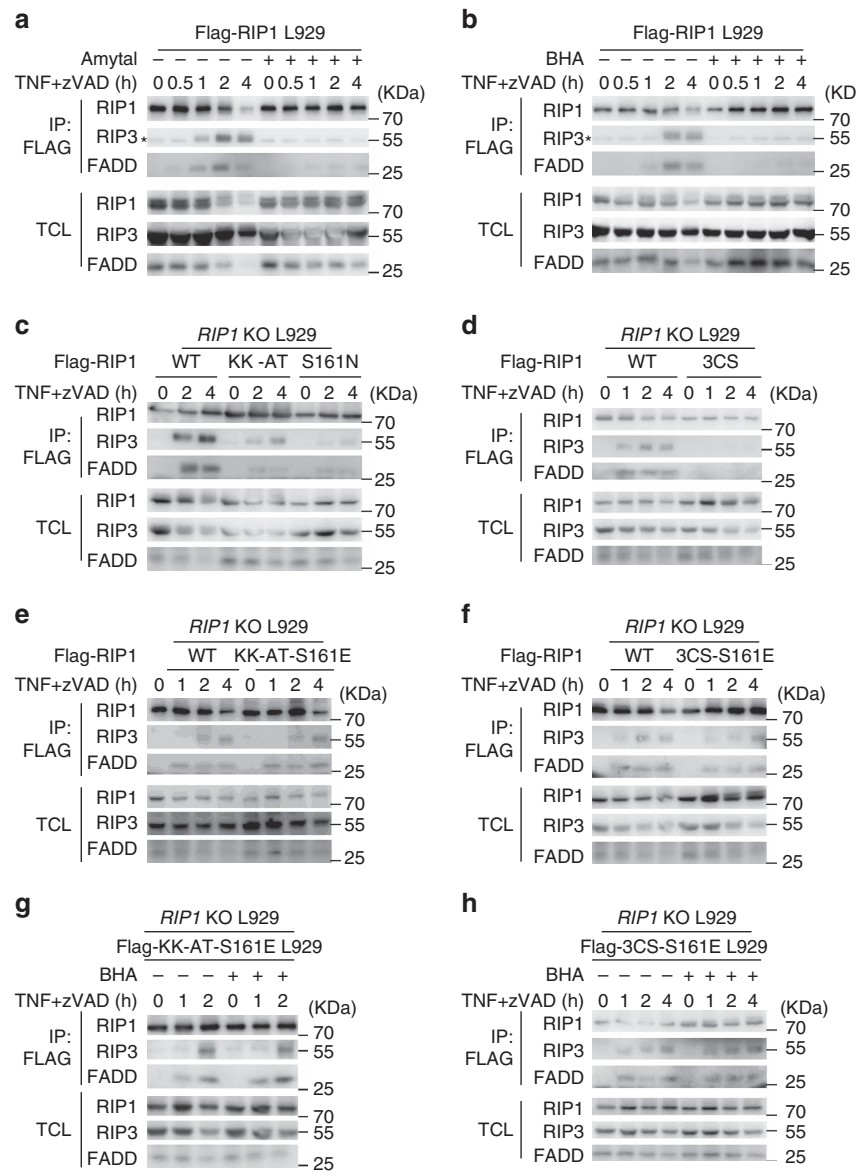

**Figure 6 | TNF-induced ROS enhance necrosome formation in a RIP1 S161 phosphorylation-dependent manner.** (**a**,**b**) Flag-RIP1 knock-in L929 cells were treated with mTNF + zVAD for the indicated time periods with/without BHA/amytal. Cell lysates were subjected to immunoprecipitation with mouse anti-Flag M2 beads and then western blotting with anti-RIP1, anti-RIP3 and anti-FADD antibodies as indicated. TCL: total cell lysate; *: non-specific band. (**c**–**f**) RIP1 KO L929 cells were infected with lentivirus encoding RIP1 WT, KK-AT, S161N, 3CS, KK-AT-S161E or 3CS-S161E for 24 h, and treated and analysed as in (**a**). (**g**,**h**) RIP1 KO L929 cells expressing RIP1 KK-AT-S161E or 3CS-S161E were treated with mTNF + zVAD for different periods of time with or without the presence of BHA. Cells were then analysed as in (**a**). Data shown in (**a**–**h**) are representatives of two to three independent experiments. See also Supplementary Figs 6,7.

RIP1 cannot interact with other proteins in necrosome, we compared WT, KK-AT and KK-AT-S161E RIP1 in their interaction with RIP3, FADD or RIP1 itself by co-expressing them in RIP1 KO HEK293 T cell line and performing co-immunoprecipitation assay. These mutations did not alter the interaction of RIP1 with other proteins (Supplementary Fig. 7a–c). We also compared the interaction of WT and KK-AT RIP1 with several ubiquitination-related proteins including cIAP1, CYLD, xIAP, MIB1, ITCH and RNF31, which were reported to regulate RIP1 in complex formation. We found that RIP1 interacted with some of these enzymes and KK-AT mutation had no effect on these interactions (Supplementary Fig. 7d). Thus, how S161 phosphorylation of RIP1 affects the recruitment of RIP3 to necrosome might not be as simple as the change of RIP1's affinities with necrosome-associated proteins.

## Discussion

Published data have suggested that the involvement of ROS in necroptosis is cell type-dependent[7,23,25,29–31]. Our data support this conclusion. It was known that TNF-induced ROS production is RIP3 dependent[8], and ROS function in a positive feedback loop to enhance necrosome formation[22]. We discovered in this study that ROS promote RIP1 autophosphorylation on S161 and cysteine 257, 268 and 586 in RIP1 are required for ROS to regulate RIP1. These three cysteines are required for the formation of disulfide bond-linked high molecular weight aggregate of RIP1 to enhance RIP1 S161 autophosphorylation, which then facilitates necrosome formation. A proposed model is shown in Fig. 7d. Since ROS induction is downstream of RIP3 activation, there could be other mechanism(s) to initiate autophosphorylation of RIP1 in the early signalling stage. As

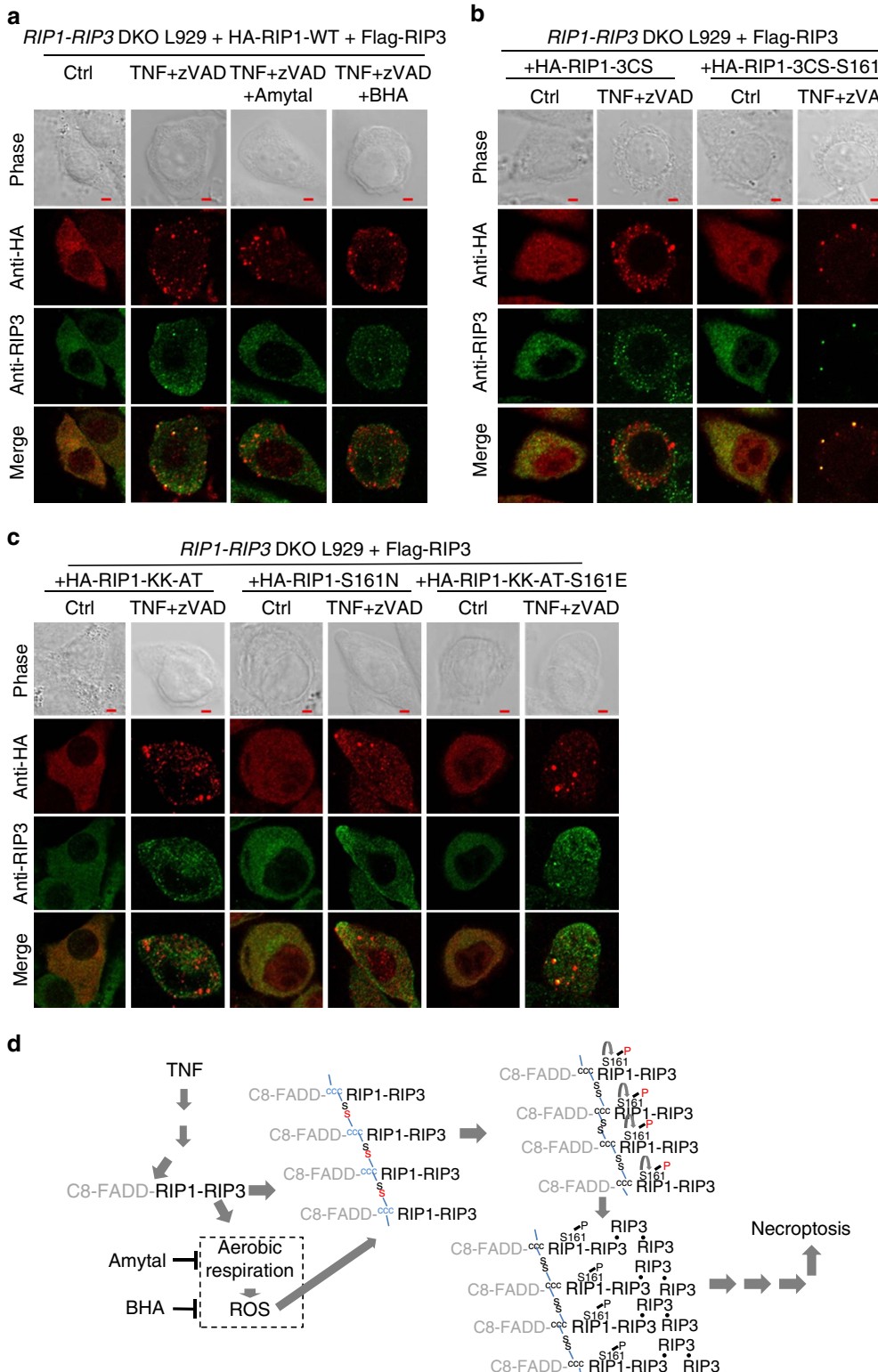

**Figure 7 | ROS as well as S161 phosphorylation of RIP1 facilitate RIP1-RIP3 co-localization in the cells undergoing TNF-induced necroptosis.** (**a**) *RIP1 and RIP3* DKO L929 cells reconstituted with HA-RIP1-WT and Flag-RIP3 were treated with mTNF + zVAD with or without the presence of BHA/amytal. Two hours after TNF stimulation, cells were fixed and immunostained for HA and RIP3 simultaneously and then subjected to confocal microscopy. (**b,c**) *RIP1 and RIP3* DKO L929 cells reconstituted with Flag-RIP3 were then reconstituted with HA-RIP1 3CS, 3CS-S161E, KK-AT, S161N, KK-AT-S161E. The cells were treated with or without mTNF + zVAD for 2 h, and then fixed and immunostained for HA and RIP3 simultaneously. (**d**) A proposed model for how ROS lead to RIP1 autophosphorylation, which enhances TNF-induced necroptosis. Scale bar: 3 μm. The images are representatives of pictures taken from at least 10 fields. See also Supplementary Figs 6,7.

kinase-dead RIP1 can still mediate a low level of cell death, non-phosphorylated RIP1 should be able to recruit a small amount of RIP3. Therefore other mechanism(s) may not be necessary because even a small amount of RIP3-dpendent ROS should be capable of initiating a positive feedback loop on RIP1 to promote S161 phosphorylation. Our data demonstrated that S161 is at least the major if not the only functional phosphorylation site in RIP1 during necroptosis and the S161 phosphorylation enables RIP1 to efficiently recruit more RIP3. Published data already showed that autophosphorylation of RIP3 occurs in necrosomes containing RIP3 homo-oligomers, which recruit and phosphorylate MLKL, and finally, the phosphorylated MLKL executes necroptosis.

RIP3 in necrosome is pro-necroptosis but caspase-8 in this signalling complex is anti-necroptosis. The role of RIP1 in this complex could be pro-apoptotic or pro-necroptotic depending on whether it only recruits caspase-8 via FADD or it also efficiently recruits RIP3 (ref. 4). The positive feedback on RIP1 phosphorylation by RIP3-dependent ROS is crucial for balancing the signalling toward necroptosis in cells like L929. And RIP1 is most likely to be the sole target of ROS in necroptosis, since KO of A20, CYLD, TRAF2, USP13, TRADD, NEMO, IKKβ, Caspase-8 or FADD in L929 cells did not affect BHA- or amytal-mediated inhibition of TNF + zVAD-induced necroptosis. We believe that S161 phosphorylation of RIP1 should be a common event in RIP1-mediated necroptosis regardless of whether ROS are involved or not, and there should be other mechanism(s) that promote(s) S161 phosphorylation. We also noticed that RIP1 3CS mutant still mediated TNF-induced necroptosis to a level higher than that mediated by kinase dead mutants (Figs 2b and 3b), supporting the idea that ROS is not the sole regulator of RIP1 autophosphorylation. Moreover, RIP1 kinase activity may have additional function in promoting necroptosis since we noticed that TNF-induced cell death in RIP1 KO cells carrying KK-AT-S161E RIP1 was not as quick as that in cells carrying S161E RIP1 (Fig. 4d). Multiple factors, including Caspase-8 inactivation/inhibition, RIP3 level in cells and ROS-mediated RIP1 S161 phosphorylation, are determinants of necroptosis, but not all of these factors are necessary for the function of necrosome in certain cell types. An example is that mitochondrial ROS is not involved in TNF-induced necroptosis in HT-29, 3T3-SA and SEVC cells[7,29].

Using Parkin-induced mitochondrial depletion, Tait et al. found that elimination of mitochondria had no effect on TNF + zVAD-induced necroptosis in a couple of cell lines[29]. Their data do not conflict with the conclusion that the ROS involvement in necroptosis is cell context-dependent. The underlying mechanism of this cell type dependence awaits further investigation. The updated data agree that ROS do not cause cell death directly, but by regulating necroptotic pathway. A number of reports showed that RIP1 kinase activity is required for apoptosis[42–44]. Since TNF can induce RIP1-dependent and -independent apoptosis, whether S161 autophosphorylation is required for TNF-induced RIP1-dependent apoptosis awaits further investigation.

It was reported recently that activated IKKα/IKKβ can inhibit RIP1 translocation from TNF receptor complex (complex I) to complex II in TNF-induced apoptosis by phosphorylating RIP1 in complex I[43]. By using IKKα/IKKβ inhibitor TPCA-1 and TAK1 inhibitor (5Z)-7-Oxozeaenol or RIP1-TAK1 double KO (RIP1-TAK1 DKO) L929 cells that were sequentially reconstituted with TAK1 and RIP1 mutants, we found that RIP1 phosphorylation by IKKα/IKKβ inhibited TNF + zVAD-induced necroptosis (Supplementary Fig. 8a–c), similar to what was observed in TNF-induced apoptosis[43]. Interestingly, the effect of TAK1-IKKα/IKKβ inhibition on necroptosis was eliminated in the cells expressing KK-AT, KK-AT-S161E or S161E RIP1

(Supplementary Fig. 8a–c). And we knew that TNF-induced complex I formation was not affected by kinase-dead or S161 phosphomimetic mutation of RIP1 (Supplementary Fig. 6b). Since autophosphorylation is mediated by RIP1 kinase activity and the common consequence of both KK-AT and S161E is diminished S161 phosphorylation, IKKα/IKKβ-mediated phosphorylation of RIP1 most likely inhibits RIP1 autophosphorylation. KK-AT mutant cannot autophosphorylate and thus cannot be regulated by IKKα/IKKβ. S161E mimics the phosphorylated form of RIP1 and thus overcomes inhibitory effect of IKKα/IKKβ on RIP1-mediated necroptosis. There are many phosphorylation sites in RIP1, including the autophosphorylation sites and those targeted by IKKα/IKKβ. Some low-mobility forms of RIP1 were showed to be phosphorylated, and band-shift of RIP1 on electrophoresis gels was widely used as an indicator of RIP1 phosphorylation[43,45]. But S161 phosphorylation contributes very little if there is any to the mobility changes of RIP1.

In conclusion, our study uncovered the mechanism of RIP3-dependent ROS in promoting necroptosis and identified key residues on RIP1 to sense redox signals. Our data provided new insight into the role of RIP1 S161 autophosphorylation in controlling the proper formation of necrosome. We believe that cell fate is determined by the balance of multiple factors in necrosome and we have demonstrated RIP1 S161 phosphorylation as an essential pro-necroptotic event.

## Methods

**Cell lines and cell culture.** Mouse fibrosarcoma L929, HeLa, human embryonic kidney (HEK) 293T, U937 and HT-29 cells were obtained from ATCC (Manassas, VA, USA). The A cell line is the NIH3T3 cell line obtained from ATCC. The N cell line is an NIH3T3 cell line obtained from Dr Sabine Adam-Klages at Universitat Kiel, Germany. This NIH3T3 cell line originated from the same cell line as A cells and was shown to undergo caspase-independent cell death in response to TNF[8]. Primary peritoneal macrophages elicited by thioglycollate medium were generated from wildtype C57BL/6 mice at the age of 8–12 weeks. MEF cell lines were isolated from wildtype mouse embryos at 13.5 dpc (days post conception). RIP1 KO HEK293T, as well as all of the gene KO L929 cell lines, was constructed by TALEN or CRISPR/Cas9 method. The target site for RIP1 was designed as 'CTACTACAT GGCGCCCGAGC', the target site for TNFR1 was 'GCTTCAACGGCACCG TGAC', the target site for RIP3 was designed as 'CTAACATTCTGCTGGA'. For MLKL, the target sites was designed as 'ATCATTGGAATACCGT'[15,46,47], and the KO cell lines were verified by sequencing of the targeted loci. RIP1 of L929 was knocked in with a Flag tag at its N-terminal by homologous recombination with the sequence of 'DYKDDDDK'. U937 was kept in ATCC-formulated RPMI-1640 medium, while others were maintained in DMEM, supplemented with 10% fetal bovine serum, 2 mM l-glutamine, non-essential amino acids to a final concentration of 0.1 mM, 100 IU penicillin and 100 mg/ml streptomycin, at 37 °C in humidified incubator with 5% CO₂. All the cell lines were authenticated by morphology and DNA sequencing, and were repeatedly tested to be mycoplasma-free as judged by the MycoAlert Mycoplasma Detection Kit (Lonza, LT-07).

**Reagents and antibodies.** Alexa Fluor 647 Goat Anti-mouse IgG (H + L) secondary antibody (1:100, A-21236) and Alexa Fluor 488 Goat Anti-rabbit IgG (H + L) Antibody (1:100, A11034) were purchased from Invitrogen (Carlsbad, CA USA). Propidium iodide (PI), BHA, 4-hydroxytamoxifen (4-OHT), CCCP and 5Z-7-Oxozeaenol were from Sigma (St. Louis, MO, USA). TPCA-1 was from Tocris Bioscience (Ellisville, MO, USA). Benzyloxycarbonyl-Val-Ala-Aspfluoromethylk-etone (zVAD) was from Calbiochem (San Diego, CA, USA). Mouse and human TNFα were from eBioscience (San Diego, CA, USA). Amytal sodium was from Marathonpharma (Northbrook, IL, USA). Mouse anti-HA (F-7), mouse anti-GAPDH (6C5), Tak1 (M-579), TRADD (H-278), TRAF2 (C-20), rabbit anti-Tom20 (FL-145) antibodies were purchased from Santa Cruz Biotechnology, Inc. (Dallas, Texas, USA), and used at a concentration of 1:500. β-Tubulin (1:1,000, 10B1) Mouse mAb was from Abmart (Berkeley Heights, NJ, USA). TNF-R1 (1:500, D3I7K) Rabbit mAb (Rodent Specific) were purchased from Cell Signaling Technology, Inc. (Danvers, MA, USA). Mouse anti-FLAG M2 and mouse anti-HA beads, mouse anti-FLAG (M2) antibodies were purchased from Sigma. Mouse anti-RIP1 antibody (1:1,000, 610459) was obtained from BD Biosciences (San Jose, CA, USA). Rabbit anti-mRIP3 (1:1,000) and rabbit anti-mFADD (1:1,000) antibodies were raised using E. coli-expressed GST-RIP3 (287-387 amino acid) and GST-FADD (full length), respectively[8,15]. Rabbit anti-human RIP3 (hRIP3) antibodies (1:1,000, ab72106) were purchased from Abcam (Cambridge Cambridgeshire, UK). Human RIP1 antibody (1:1,000) was raised in rabbit using amino acids 555-671. For Flag-tagged mTNF preparation, matured mouse TNFα

(a truncated form of prematured mouse TNFα from 77–235a.a.) was cloned into pET28a vector between NcoI and XhoI enzymatic sites with an N-terminal fused 3xFlag tag. Protein expression in BL21de3 was carried out overnight at 30 °C in auto-induction medium. Then the lysates were first purified by ammonium sulfate precipitation. Soluble fraction from 30–65% saturation of ammonium sulfate was further purified with M2 beads. After eluted with 3xFLAG peptide (Sigma), proteins were finally purified with Superdex-75 120 ml prepacked column (GE Healthcare, Little Chalfont, UK).

**Plasmids construction.** YFP-Parkin (plasmid #23955) was purchased from Addgene (Cambridge, MA, USA). Full-lengths of *TAK1*, *RIP1*, mouse/human *RIP3*, *FADD* and UB-related genes were amplified from our cDNA library. To generate fusion proteins, *HBD\*(G521R)*, *tTNFR1*, *RIP1ΔDD*, *RIP3-RHIM^mut* and *MLKLΔPD* were amplified by standard PCR from corresponding templates. *RIP1* mutations (D138N, KK-AT, S161A, S161E, S161N etc.) were introduced by two-round PCR. Lentiviral vector used was pBOB (Addgene plasmid #12337), which had been modified by inserting DNA fragments encoding Flag/HA tag. pBOB was used as plasmid backbone for all of the constructs in this study. Exo III-assisted ligation-independent-cloning method[48] was used for all subcloning. All plasmids were verified by DNA sequencing. The details of the plasmids sequences are available upon request.

**Immunoprecipitation and western blotting.** For complex I or necrosome immunoprecipitation, cells were seeded in a 100 mm tissue culture plate, grew to reach confluency and then were treated with 10 ng/ml Flag-mTNF or mTNF + zVAD (20 μM) for different periods of time. After treatment, cells were scrapped from the plates and rinsed twice with ice-cold PBS and lysed with buffer published before[35]. Cell lysates were centrifuged at 20,000 × g for 30 min, and the supernatants were subjected to immunoprecipitation with mouse anti-Flag M2 beads and anti-HA beads at 4 °C overnight. The beads were then washed three times in lysis buffer the next day and the 1.2X Laemmli buffer eluted proteins were then analysed by routine western blotting. Uncropped scans of blots were supplied as Supplementary Fig. 9 in the supplementary information.

**Cell viability assay.** Cells were trypsinized to suspended single cells from tissue culture plates and incubated with 5 μg ml$^{-1}$ PI in PBS for 10 min. PI negative cells (indicating living cells) were analysed by FACS (BD Calibur; or Beckman Coulter EPICS XL) with excitation/emission maxima of 535/617 nm approximately.

**Measurement of oxygen consumption rate.** Oxygen consumption rates of cells were measured by the Seahorse Bioscience Extracellular Flux Analyzer (XF96; Seahorse Bioscience) according to manufacturer's protocol. Briefly, 8,000 cells (each well) were cultured overnight in custom XF96 microplates. Before measurement, cells were washed, and immersed in fresh medium for 1 h. The assay procedure was set as 'Mix-03:00, Wait-00:00, Measurement-03:00' for 10 cycles. After measurement, cells were trypsin-digested and counted for cell number. Oxygen consumption rates of cells were normalized to cell number.

**Measurement of ROS.** Cells were cultured and treated with indicated stimuli. After treatment, the cells were washed with warm PBS, removed from the plates with trypsin-EDTA, pelleted at 200 g for 3 min, re-suspended in Hank's balanced-salt solution containing 5 μM MitoSOX for 10 min and analysed by flow cytometry with excitation/emission maxima of approximately 510/580 nm.

**In vitro kinase assay.** *In vitro* kinase reaction was performed in buffer containing 40 mM Tris, pH 7.5, 20 mM MgCl$_2$, 0.1 mM Na$_3$VO$_4$, 10 μM ATP.

**Sample preparation for confocal microscopy.** Reconstituted *RIP1-RIP3* DKO L929 was seeded on circular cover glass (72222-01, Electron microscopy science, Hatfield, PA, USA) pretreated with Poly-ʟ-lysine hydrobromide (Sigma) for 15 min. After stimulated with mTNF + zVAD for 2 h, cells were washed with PBS three times, and fixed with 3% paraformaldehyde for 20 min at room temperature (RT). 0.25% Triton X-100 was then used to break cell membrane for 10 min at RT. These fixed samples were then washed with PBS and blocked with 3% BSA for 30 min at RT. After aspiration of blocking buffer, primary anti-HA (mouse, 1:100) and anti-RIP3 (Rabbit, 1:100) antibodies diluted in blocking buffer were used at 4 °C overnight. The samples were then washed five times with PBS at RT and then subjected to fluorescence-labelled secondary antibodies, goat anti-mouse Alexa-Fluor 647 and goat anti-rabbit AlexaFluor 488. This incubation should be protected from light for 60 min at RT. Cells were then washed three times with PBS and then stored in PBS, ready for confocal analysis. All images were captured and processed using identical settings in the Zeiss LSM 780 laser scanning confocal microscope with a 63 × /1.40 NA oil objective. Duplicate cultures were examined, and similar results were obtained in at least three independent experiments.

**Determination of S161 phosphorylation by mass spectrometry.** Flag-RIP1 was immunoprecipitated with M2 beads from mTNF + zVAD treated *RIP1* KO L929 cells expressing Flag-RIP1. Proteins in IP samples were precipitated with 10% TCA, and then washed three times with iced acetone. Proteins were then dissolved in 8 M urea/50 mM NH$_4$HCO$_3$, and disulfide bonds were reduced with 10 mM tris (2-carboxyethyl)phosphine (TCEP) followed by alkylation with 40 mM chloroacetamide. After 8 M urea was diluted to 2 M urea with 50 mM NH$_4$HCO$_3$, trypsin was added with a protein to trypsin ratio of 50:1, and digestions were performed at 37 °C for 12–16 h. Two synthetic phosphopeptides, pSRTPPSAPSQSR and GHLpSEGLVTK (50 fmole each), were added into the samples as internal controls for the quantitation of phosphopeptides. Tryptic peptides were desalted with C18 STAGEtip followed by phosphopeptide enrichment using immobilized metal affinity chromatography as described previously[49]. Briefly, peptides were dissolved in 50 μl 60% ACN (acetonitrile)/1% AA (acetic acid), and then 5 μl immobilized metal affinity chromatography beads were added. The tubes were rigorously shaken for half an hour. Then beads were washed with 60% CAN/1% AA whose volume was twofold of beads volume. The phosphopeptides were eluted with 6% NH$_3$H$_2$O the volume of which was threefold of beads volume.

Phosphopeptides were analysed using MS tripleTOF 5600. To quantify RIPK1 phosphopeptide containing S161, MS was operated in high-resolution MRM mode. The m/z values of the precursor ions representing RIP1 S161 and two synthetic phosphopeptides were in the inclusion list. The MS1 scan was 0.25 s and covered 350–1,250, and MS2 scan of three precursor ions was 0.5 s and covered 100–1,800, resulting in the cycle time of 1.75 s. The collision energy of each precursor ion was calculated using the equation: $|CE| = (0.044)*(m/z) + 4$ for the peptides with $+2$ charge, and $|CE| = (0.05)*m/z + 3$ for the peptides with $+3$ charge. MS data analyses were performed with Peakview (version 2.2) software.

**Identification of cysteine modification by mass spectrometry.** RIPK1 complexes were purified with M2 beads in Flag-RIP1 reconstituted L929 cells. During IP, 10 mM N-ethylmaleimide was added. RIP1 was eluted with SDS sample buffer containing no reducing agent followed by SDS-PAGE. The gels corresponding to aggregated RIP1 or monomer RIP1 were excised and cut into cubes. Proteins were further reduced with 10 mM Tris(2-carboxyethyl) phosphine hydrochloride for 20 min, followed by treatment with 30 mM iodoacetamide for 20 min. Sequencing grade soluble trypsin was used for digestion. The resulting peptides were subjected to MS analysis. The data-dependent tandem MS analysis was performed in a Triple TOF 5600 (Sciex). The MS1 scan is 0.25 s and covers 350–1,250, and 20 precursor ions are selected for MS2 analysis and covers 100–1,800, resulting in the cycle time of 1.25 s. MS data were searched with ProteinPilot Software (V.4.5).

**Statistical evaluation.** Statistical analysis was performed with Prism software (GraphPad Software). Data are represented as means ± s.e.m. Two-tailed Student's *t*-test was used to compare differences between treated groups and their paired controls. Differences in compared groups were considered statistically significant with P values lower than 0.05; *P < 0.05; **P < 0.01.

**Data availability.** All relevant data are available from the authors and data in the manuscript are available as source data.

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

## Acknowledgements

This work was supported by the National Natural Science Foundation of China (91029304, 31420103910, 31330047 and 81630042), the National Basic Research Program of China (973 Program; 2015CB553800, 2013CB944903, 2014CB541804), the 111 Project (B12001), the National Science Foundation of China for Fostering Talents in Basic Research (J1310027).

## Author contributions

Y.Z., S.S.S. and J.H. designed experiments, performed experiments and analysed the results. S.Z. and Z.Y. performed experiments. C.-Q.Z. performed the mass spectrometry experiment and analysed the obtained results. X.C. and Q.C. generated essential tools for the study. Z.-H.Y. and R.W. contributed to the plasmid preparation of the study. Y.Z. and J.H. wrote the manuscript.

## Additional information

**Competing financial interests:** The authors declare no competing financial interests.

