## [Peer Review File · Nature Communications]

Reviewers' Comments:

Reviewer #2 (Remarks to the Author)

The manuscript of Zhang et al. describes the role of RIP1 phosphorylation in the necrosome assemblage. Taking into account that the role of RIP1 in the necroptosis process is already well established and that ROS generation promotes stabilization of RIP1/RIP3 necrosome complex, the present work attempt to demonstrate that cysteine modification on RIP1 was responsible for the following autophosphorylation of RIP1.

The data reported are really enormous and in some instance they results superfluous rendering the manuscript hard reading.

For example, the data reported in supplementary Table 1 are not necessary, because the aim is to dissect the role of ROS on RIP1 and necrosome formation. Moreover, there is no explanation for the selection of that genes and no real evidence of the gene KO were reported but just cell death was indicated. Same concern for the data reported in Supplementary Fig.1 were different time points were used in order to demonstrated the similar or opposite cell response to treatments. The authors premise and the literature reports said that the conditions used for induction of necroptosis are not common. Also in this case only cell survival is reported.

Fig1. Panel a, the decrement in TOM20 reported cannot be the only evidence to ascertain mitochondria depletion under Parkin transfection. Therefore, the following results on cell viability can be due to a mixed population of cells with different mitochondria content. Lack of adequate controls is indeed the main concerns related to the manuscript.

Fig.2 No data were reported on the dimerization/oligomerization system used, also in this case the results are just on the survival rate at very different time points. The data should demonstrated the occurrence of the dimerization/oligomerization process.

The subsequent results could be important if supported by the right controls and with regards to cysteine oxidation it will be important to demonstrated the ROS-mediated oxidation (by mass spectrometry ?) beside overexpressing the mutated forms.

Reviewer #3 (Remarks to the Author)

In this study, Zhang et al studied the potential mechanism of ROS-promoted, RIP1-mediated necroptosis. Based on their study, the authors concluded that ROS promotes necroptosis through inducing RIP1 autophosphorylation in L929 cells. The authors demonstrate that ROS activates RIP1 autophosphorylation at S161 and thus enables RIP1/ RIP3 necrosome formation to induce necroptosis. The data presented in this study are generally sound and well controlled. The findings of this study is interesting and important for general readers interested in cell biology, particularly in the research of cell death.

Some specific points need to be addressed before it is published.

1. Since most of the experiments were done in L929 cells, to generalize the findings, some key experiments need to be verified in an additional type of cells, such as MEF cells. For instance, Do S161 and 3CS1 mutations of RIP1 have the similar effect on necroptosis in MEF cells?.
2. Does the mutation of 3 cysteines (3CS1) in RIP1 affect its autophosphorylation of S161 during necroptosis?
3. The confocal images in Fig.6 is too dark, especially for the Red images.

Reviewer #4 (Remarks to the Author)

The manuscript by Zhang and colleagues reports on the role of mitochondrial ROS production and RIP1 auto-phosphorylation on S161 in TNF-mediated necroptosis. The kinase activity of RIP1 is

known to be required for necroptosis induction, but the mechanism by which RIP1-mediated phosphorylation transmits the death signal is currently unknown. Similarly, ROS scavenging was previously reported to protect cells from TNF-induced necroptosis by preventing necrosome formation, but the link between mitochondrial ROS production and RIPK1 activation is currently unclear. In this study, the authors provide evidences suggesting that cysteine residues in RIP1 sense ROS, which would lead to the activation of its kinase activity. Active RIP1 would then lead to auto-phosphorylation on S161, which would be sufficient for TNF-mediated necrosome formation and necroptosis induction.

The results provided by the authors are very interesting and could fill in some of the gaps in our current understanding of the molecular mechanism regulating TNF-mediated necroptosis. Nevertheless, some of the findings seem correlative and lack clear demonstration. For instance, additional experiments directly linking RIP1 cysteines/ROS sensing and ROS sensing/RIP1 kinase activation would be required (use of recombinant proteins?). The authors say that ROS is sensed by cysteines and function to enhance RIPK1 kinase activity but this is not formally demonstrated. There is also no data showing that auto-phosphorylation of RIP1 is affected in the 3CS mutant. In addition, it is unclear whether the effect observed with the 3CS mutant originates from a general folding defect of RIP1. Is this mutant still recruited to complex I? Importantly, the authors should show that adding the S161E mutation to the 3CS mutant rescues necrosome formation and necroptosis induction (which can not be prevented by Amytal or BHA), ideally both following TNF+ZVAD stimulation and by using the oligomerization strategy. Similarly, although the authors convincingly show that auto-phosphorylation of RIP1 on S161 bypass the requirement of RIP1 kinase activity for necrosome assembly and necroptosis induction, the direct molecular consequence of this phosphorylation event is still very vague. What is it actually doing? The results presented in supplemental Figure 5C-E lack proper controls, the RIP1 signal could originate from non-specific binding to the beads. The authors should try other approaches (use of recombinant proteins?) to answer this problem.

Additional specific comments:

- Contrary to the authors' statement, the requirement of RIP1 kinase activity for apoptosis induction has been well documented and demonstrated both in vitro and in vivo (Biton and Ashkenazi Cell 2011; Dondelinger et al. CDD 2013, Abhari et al. Oncogene 2013, Dondelinger et al. Mol Cell 2015, Boutafalla et al. CDD 2015, Kondylis et al. Cancer Cell 2015, Vlantis et al. Immunity 2016). The involvement of RIP1 enzymatic activity in TNF-mediated apoptosis was shown to depend on the nature of the trigger, due to the existence of different cell death checkpoints (Ting and Bertrand, Trends Immunol 2016). In addition, ROS scavenging was demonstrated to protect cells from TNF-induced RIP1-mediated apoptosis by acting upstream of complex Iib formation (Dondelinger et al. CDD 2013). It is therefore tempting to speculate that ROS and the kinase activity of RIP1 regulate apoptosis and necroptosis via a common mechanism. The authors should therefore test whether phosphorylation of RIP1 on S161 is also sufficient to trigger TNF-mediated RIP1-dependent apoptosis, and whether the 3CS mutant is protected from apoptosis. This is of great relevance for the understanding of the molecular consequence of this phosphorylation event (adaptation of Figure 7D?). The authors claim that "KK-AT RIP1 showed no difference from cells reconstituted with WT RIP1 in TNF-induced apoptosis" but the trigger used is not specified. As shown in Supplemental Figure 1, ROS scavenging protects different cell types from TNF-induced necroptosis, so other cellular systems can even be used.

- For all the graphs, the results should be pooled and statistical analysis performed. For proper interpretation of the data, the timing used in the different experiments should stay consistent (up to 4/6h for all).

- Molecular weight markers should be added to all the blots.

- In Figure 2, the authors use an oligomerization strategy to induce necroptosis, but the system is

not well characterized. For instance, does necroptosis induced by RIP1deltaDD-HB rely on RIPK3 and MLKL?

- In Figure 3C, the authors show cell death induced by RIP1deltaDD-HBD-KK-AT. Is this cell death RIPK3 and MLKL dependent?

- In Figure 3D, the authors should explain the rationale for having an additional D mutation for some but not all phosphorylated residues. RIP1 protein levels should also be shown in order to allow comparison between the different mutants.

- The results from Figure 4D are very interesting but it would be more appropriate to have a separate graph for each KD mutant, and each graph should contain the KD mutant alone, the KD mutant with S161E, and S161E alone.

- In Figure 5D, 7B and supplemental Figure 4C-D, the results for IP RIP3 should be presented with normal contrast/illumination. In Figure 5D, it even looks like if there was an intention to hide the fact that RIP3 binds RIP1 (even at 0h) in the S161N. The resolution of Figure 5E is very bad. FADD recruitment should also be analyzed in these IPs. The quality of the IPs in supplemental Figure 4C-E is not very good. How do the authors explain that ROS scavenging affects recruitment of FADD and RIP3 in supplemental Figure 4D (and C??), but that the 3CS-RIP1 mutation only affects binding to RIP3 and not to FADD, and that addition of BHA still does not affect FADD binding (Figure 7C)? Supplemental Figure 4F also seem to show that binding of RIP1 to RIP3 is not affected by BHA...

- The rationale for the results presented in the supplemental part vs article is not always clear to me. Some of the work presented in the article is a confirmation of previously published studies while very important results are put in the supplemental (such as results from supplemental Figure 4-5). The results from Figure 6 are not convincing (we barely see anything), and could easily go to the supplemental part. Also, the results presented in Figure 1, 2 and 5A-B could be combined in one Figure. The results from Figure 7 on ROS/RIP1 cysteines would better fit with the first part of the story on ROS scavenging than at the end of the article.

- The authors should already refer in the introduction to the work showing that ROS promote TNF-mediated RIP1-dependent apoptosis and necroptosis by acting downstream of complex I and upstream of complex II and the necrosome (Dondelinger et al CDD 2013, Schenk et al Oncogene 2015). In p2 line 34, the authors define complex IIb and necrosome as the same complex. However, complex IIb was initially defined as a RIP1-dependent caspase-8 activating complex II (RIPK1/FADD/caspase-8) in contrast to the TRADD/FADD/caspase-8 complex IIa (Wilson et al. Nat Immunol 2009). In order to stay consistent with original definition and to avoid confusion, I would suggest to dissociate the apoptotic complex IIb from the necrosome. The authors should also check the relevance of the literature they are citing. For example, the work cited in reference to "it is also clear that RIP1 kinase activity is involved in necroptosis"¹⁷⁻²⁰ (p2 line 43) is certainly not the most logical to me.

Reviewers' comments:

Reviewer #1 (Remarks to the Author):

The manuscript of Zhang et al. describes the role of RIP1 phosphorylation in the necrosome assemblage. Taking into account that the role of RIP1 in the necroptosis process is already well established and that ROS generation promotes stabilization of RIP1/RIP3 necrosome complex, the present work attempt to demonstrate that cysteine modification on RIP1 was responsible for the following autophosphorylation of RIP1.

The data reported are really enormous and in some instance they results superfluous rendering the manuscript hard reading.

For example, the data reported in supplementary Table 1 are not necessary, because the aim is to dissect the role of ROS on RIP1 and necrosome formation. Moreover, there is no explanation for the selection of that genes and no real evidence of the gene KO were reported but just cell death was indicated.

Response: *The data in Table 1 excluded a number of signaling components in necroptosis pathway as targets of ROS. The genes we selected are those involved in TNF-induced necroptosis. All gene knockouts were confirmed by DNA sequencing of corresponding genes. We agree with the reviewer that supplementary table 1 might not be necessary and removed it in the revised manuscript but briefly mentioned related results in the discussion section.*

Same concern for the data reported in Supplementary Fig.1 were different time points were used in order to demonstrated the similar or opposite cell response to treatments. The authors premise and the literature reports said that the conditions used for induction of necroptosis are not common. Also in this case only cell survival is reported.

Response: *The reviewer is right that due to different sensitivities to necroptosis stimuli, different lengths of time course have to be used for different cells. To make it easy to understand, reaching 60-80% cell death was used to determine the last time point of treatment to be used for each different cell/cell lines. We explained this in the revised manuscript (figure legend of figure S1d). We also included ROS data in the revised manuscript (Fig. S1e).*

Fig1. Panel a, the decrement in TOM20 reported cannot be the only evidence to ascertain mitochondria depletion under Parkin transfection. Therefore, the following results on cell viability can be due to a mixed population of cells with different mitochondria content. Lack of adequate controls is indeed the main concerns related to the manuscript.

Response: *We agree and included oxygen consumption and MitoTracker Red staining in the revised manuscript (Fig. S1b, S1c). Based on MitoTracker Red staining, the depletion of mitochondria is relatively uniform among cells.*

Fig.2 No data were reported on the dimerization/oligomerization system used, also in this case the results are just on the survival rate at very different time points. The data should demonstrated the occurrence of the dimerization/oligomerization process.

Response: *We agree. We included the data of 4-OHT induced oligomerization of HBD* fused tTNFR1, RIP1, and RIP3 in revised manuscript (Fig. S1f, S1g, S1h). The oligomerization of MLKLΔPD-HBD* was described in our previous publication and we cited it in the revised manuscript.*

The subsequent results could be important if supported by the right controls and with regards to cysteine oxidation it will be important to demonstrated the ROS-mediated oxidation (by mass spectrometry ?) beside overexpressing the mutated forms.

Response: *We agree that this is a very important question and performed experiments as suggested. We detected oxidation-dependent large RIP1 containing aggregates in TNF-treated L929 cells and identified by MS that C257, 268, and 586 residues in the aggregates were oxidized (Fig. 2a, 2d, 2e, S2c in revised manuscript).*

Reviewer #2 (Remarks to the Author):

In this study, Zhang et al studied the potential mechanism of ROS-promoted, RIP1-mediated necroptosis. Based on their study, the authors concluded that ROS promotes necroptosis through inducing RIP1 autophosphorylation in L929 cells. The authors demonstrate that ROS activates RIP1 autophosphorylation at S161 and thus enables RIP1/ RIP3 necrosome formation to induce necroptosis. The data presented in this study are generally sound and well controlled. The findings of this study is interesting and important for general readers interested in cell biology, particularly in the research of cell death.

Some specific points need to be addressed before it is published.

1. Since most of the experiments were done in L929 cells, to generalize the findings, some key experiments need to be verified in an additional type of cells, such as MEF cells. For instance, Do S161 and 3CS1 mutations of RIP1 have the similar effect on necroptosis in MEF cells ?

Response: *We performed the suggested experiments. The MEF data are included as Fig. S5b in the revised manuscript, which support our conclusion.*

2. Does the mutation of 3 cysteines (3CS1) in RIP1 affect its autophosphorylation of S161 during necroptosis?

Response: *This is an excellent question. We performed this experiment. 3CS mutation*

prevents S161 autophosphorylation during necroptosis (Fig. 5a in revised manuscript).

3. The confocal images in Fig.6 is too dark, especially for the Red images.

Response: *The reviewer is right. We have improved our images in the revised manuscript (Fig. 7a-c).*

Reviewer #3 (Remarks to the Author):

The manuscript by Zhang and colleagues reports on the role of mitochondrial ROS production and RIP1 auto-phosphorylation on S161 in TNF-mediated necroptosis. The kinase activity of RIP1 is known to be required for necroptosis induction, but the mechanism by which RIP1-mediated phosphorylation transmits the death signal is currently unknown. Similarly, ROS scavenging was previously reported to protect cells from TNF-induced necroptosis by preventing necrosome formation, but the link between mitochondrial ROS production and RIPK1 activation is currently unclear. In this study, the authors provide evidences suggesting that cysteine residues in RIP1 sense ROS, which would lead to the activation of its kinase activity. Active RIP1 would then lead to auto-phosphorylation on S161, which would be sufficient for TNF-mediated necrosome formation and necroptosis induction.

The results provided by the authors are very interesting and could fill in some of the gaps in our current understanding of the molecular mechanism regulating TNF-mediated necroptosis. Nevertheless, some of the findings seem correlative and lack clear demonstration. For instance, additional experiments directly linking RIP1 cysteines/ROS sensing and ROS sensing/RIP1 kinase activation would be required (use of recombinant proteins?). The authors say that ROS is sensed by cysteines and function to enhance RIPK1 kinase activity but this is not formally demonstrated. There is also no data showing that auto-phosphorylation of RIP1 is affected in the 3CS mutant.

Response: *The reviewer is right and we performed experiments as the reviewer suggested. We showed in Fig. 5a of the revised manuscript that 3CS mutation in RIP1 blocked S161 phosphorylation, indicating that the cysteine oxidation is required for RIP1 auto-phosphorylation on S161 during necroptosis. To show directly that the cysteine oxidation positively affects S161 auto-phosphorylation, we need to know what kind of oxidation occurred on cysteines. We found that TNF induced high molecular weight RIP1-containing complex/aggregate, which depends on disulfide bound as β -mercaptoethanol can disrupt it (Fig. 2a in the revised manuscript). This oxidation-dependent complex cannot be formed by RIP1 3CS mutant (Fig. 2d), indicating the formation of intermolecular disulfide bounds by these cysteines. MS analysis also detected oxidation of C257, 268, and 586 in the high molecular complex (Fig. 2e, S2c in revised manuscript). These data indicated that oxidation of these three cysteines was responsible for the formation of disulfide bounds in the high molecular*

weight RIP1-containing complex, and the formation of this complex promotes S161 phosphorylation.

We also followed the reviewer's suggestion to use recombinant proteins. Since it is difficult to prepare active recombinant protein of RIP1 from *E. coli*, we expressed and purified RIP1, RIP1 KK-AT and RIP1 3CS mutant proteins using 293T cells. We found that all of these three proteins formed disulfide bond-linked complexes during the purification (Fig. 5d). We used DTT to break the disulfide bonds in these complexes and H₂O₂ to enhance the oxidation of these complexes before in vitro kinase assay. Since there is a high non-specific background of the in vitro kinase assay using these immunopurified RIP1 proteins, we cannot use conventional ³²P labelling method to detect S161 autophosphorylation of RIP1. Therefore, we used MS to quantitate S161 phosphorylation levels before and after in vitro kinase reactions. We detected background S161 phosphorylation in WT RIP1 but not in KK-AT and 3CS (Fig. 5e), which confirmed that S161 phosphorylation is autophosphorylation. In vitro incubation with ATP increased S161 phosphorylation of WT RIP1 but not 3CS RIP1, indicating that the disulfide bonds by these three specific cysteines but not by other cysteines are functional in regulating RIP1 kinase activity. H₂O₂ enhanced in vitro autophosphorylation of WT RIP1 but not KK-AT or 3CS, and DTT destroyed WT RIP1 S161 autophosphorylation activity. These data support the notion that the three cysteines mediate complex formation of RIP1 and promote RIP1 autophosphorylation on S161. The background S161 phosphorylation of WT RIP1 most likely occurred during protein purification since there should be some traced ATP around and the three cysteines should participate in the formation of some of the complexes during purification.

In addition, it is unclear whether the effect observed with the 3CS mutant originates from a general folding defect of RIP1. Is this mutant still recruited to complex I? Importantly, the authors should show that adding the S161E mutation to the 3CS mutant rescues necrosome formation and necroptosis induction (which cannot be prevented by Amytal or BHA), ideally both following TNF+ZVAD stimulation and by using the oligomerization strategy.

Response: The reviewer is right that we should be cautious on possible global structural change induced by mutations. As suggested, we examined whether 3CS was recruited to complex I and found it was in complex I with a comparable level as WT RIP1 (Fig. S6a in revised manuscript). We added S161E mutation to the 3CS mutant and found it can rescue necroptosis induction and necrosome formation (Fig. 5b, 6f in revised manuscript). The 3CS-S161E mutant was not sensitive to BHA (Fig. 5b, 6h in revised manuscript). S161E mutation also bypassed 3CS effect in RIP1 Δ DD-HBD* oligomerization-induced cell death (Fig. 5c in revised manuscript). These data also suggest that 3CS mutant has no folding defect that can affect its function in necroptosis.

Similarly, although the authors convincingly show that auto-phosphorylation of RIP1 on S161 bypass the requirement of RIP1 kinase activity for necrosome assembly and

necroptosis induction, the direct molecular consequence of this phosphorylation event is still very vague. What is it actually doing? The results presented in supplemental Figure 5C-E lack proper controls, the RIP1 signal could originate from non-specific binding to the beads. The authors should try other approaches (use of recombinant proteins?) to answer this problem.

Response: *We showed that S161E bypassed the defect of kinase dead mutation in necrosome assembly, and in the revised manuscript we showed that S161E also bypassed 3CS mutation (Fig. 5b,6f). It is clear that S161 phosphorylation had no role in the formation of disulfide bond-linked high molecular weight complex since TNF+zVAD did not induce this complex formation in 3CS-S161E expressing cells (Fig. 5f). This is predictable because the three cysteines were mutated. Since S161 phosphorylation clearly facilitates the formation of necrosome (Fig. 6e,6f), it is likely that S161E helps RIP1 to interact with other component(s) more efficiently. However, we were unable to detect any difference in these interactions using co-expression assay (Fig. S5C-E in our old version of the manuscript). The reviewer is right that we missed controls to exclude the possibility that RIP1 signal was originated from non-specific binding to the beads. We included proper controls in the new experiments and reached the same conclusion (Fig. S7a-c in the revised manuscript). We followed the reviewer's suggestion to use recombinant proteins isolated from 293T cells. The homo-interaction of RIP1, and hetero-interaction of RIP1 and RIP3 were not affected by S161E mutation in in vitro pulldown assay. Since it would not bring in any new information, we did not include this data in the revised manuscript. At this stage, we do not know how the S161 phosphorylated RIP1 enhances the formation of necrosome.*

Additional specific comments:

- Contrary to the authors' statement, the requirement of RIP1 kinase activity for apoptosis induction has been well documented and demonstrated both in vitro and in vivo (Biton and Ashkenazi Cell 2011; Dondelinger et al. CDD 2013, Abhari et al. Oncogene 2013, Dondelinger et al. Mol Cell 2015, Boutaffala et al. CDD 2015, Kondylis et al. Cancer Cell 2015, Vlantis et al. Immunity 2016). The involvement of RIP1 enzymatic activity in TNF-mediated apoptosis was shown to depend on the nature of the trigger, due to the existence of different cell death checkpoints (Ting and Bertrand, Trends Immunol 2016). In addition, ROS scavenging was demonstrated to protect cells from TNF-induced RIP1-mediated apoptosis by acting upstream of complex IIb formation (Dondelinger et al. CDD 2013). It is therefore tempting to speculate that ROS and the kinase activity of RIP1 regulate apoptosis and necroptosis via a common mechanism. The authors should therefore test whether phosphorylation of RIP1 on S161 is also sufficient to trigger TNF-mediated RIP1-dependent apoptosis, and whether the 3CS mutant is protected from apoptosis. This is of great relevance for the understanding of the molecular consequence of this phosphorylation event (adaptation of Figure 7D?). The authors claim that "KK-AT RIP1 showed no difference from cells reconstituted with WT RIP1 in TNF-induced

apoptosis” but the trigger used is not specified. As shown in Supplemental Figure 1, ROS scavenging protects different cell types from TNF-induced necroptosis, so other cellular systems can even be used.

Response: The reviewer is right that a number of reports showed that RIP1 kinase activity is required for apoptosis. We re-examined our data and performed more experiments to address whether S161E and 3CS mutation affects apoptosis in TNF-treated L929 cells. We still used RIP1-RIP3 double knockout (KO) L929 cells reconstituted with vector, WT or different RIP1 mutant to address this question. As shown below, TNF can efficiently induce cell death in RIP1-RIP3 DKO L929 cells. Because the cell death can be blocked by zVAD, it should be apoptosis. The induction of apoptosis in RIP1-RIP3 DKO cells was predictable because it is known that TNF can induce TRADD-dependent apoptosis when RIP1 was deleted or knocked down (PMID:16611992; PMID:22089168). When RIP1 was reconstituted in RIP1-RIP3 DKO cells, RIP1-dependent apoptosis should occur. We detected TNF-induced apoptosis in WT RIP1-reconstituted cells but the level of cell death was slightly lower than vector cells (see figure below). In support of the participation of RIP1 S161 phosphorylation in apoptosis, more cell death was observed in S161E-reconstituted cells. Apoptosis still occurred in KK-AT and 3CS cells treated with TNF, but cell death was less than WT reconstituted cells. The KK-AT-S161E and 3CS-S161E cells were more sensitive to TNF-induced apoptosis than KK-AT and 3CS cells. Overall, the differences of apoptosis in those reconstituted cell lines are not significant enough to draw a conclusion. It is possible that the presence of RIP1-independent apoptosis reduced or interfered with RIP1-dependent apoptosis in our system.

Since TNF+Smac memetic-induced apoptosis is RIP1 dependent and TRADD-independent (PMID:18485876), we think deleting TRADD gene in RIP1-RIP3 DKO L929 cells and then repeating the experiments shown in the figure above should be able to unambiguously show whether S161 phosphorylation plays a role in apoptosis. Unfortunately, we failed to generate the triple KO cell line so far.

Since overexpression of RIP1 can cause necroptosis in the presence of RIP3 or apoptosis in the absence of RIP3, we also tested overexpression of RIP1 and its mutants in RIP1-RIP3 DKO cells. Our preliminary data is shown below. Overexpression of kinase-dead or 3CS RIP1 cannot cause apoptosis and S161E caused more cell death than WT RIP1. S161E can overcome KK-AT's effect in promoting apoptosis. Collectively, the data suggest that the reviewer's prediction -S161 phosphorylation plays a role in

TNF-induced apoptosis - could be right.

Because a systematic approach may be needed to reach a conclusion of whether ROS mediated S161 phosphorylation can promote apoptosis, we only can leave this issue for further investigation in the future. We revised our comment on this issue in the discussion section of our revised manuscript as that “A number of reports showed that RIP1 kinase activity is required for apoptosis. Since TNF can induce RIP1-dependent and -independent apoptosis, whether S161 autophosphorylation is required for TNF-induced RIP1-dependent apoptosis awaits further investigation”.

- For all the graphs, the results should be pooled and statistical analysis performed. For proper interpretation of the data, the timing used in the different experiments should stay consistent (up to 4/6h for all).

Response: *As requested by the reviewer, we performed new experiments to replace some figures with the time of treatment less than 4 hours. However, this was restricted to the experiments using L929 cells because other cell types have different sensitivities to necroptosis stimuli and different time points of treatment have to be used. We have pooled data from two or three independent experiments to perform statistical analysis as requested by the reviewer.*

- Molecular weight markers should be added to all the blots.

Response: *They were added as requested.*

- In Figure 2, the authors use an oligomerization strategy to induce necroptosis, but the system is not well characterized. For instance, does necroptosis induced by RIP1 Δ DD-HB rely on RIPK3 and MLKL?

- In Figure 3C, the authors show cell death induced by RIP1 Δ DD-HBD-KK-AT. Is this cell death RIPK3 and MLKL dependent?

Response: *We performed experiments to address these two questions (Fig. S3b-c in revised manuscript). Both RIP1 Δ DD-HBD* oligomerization- and RIP1 Δ DD-HBD*-KK-AT oligomerization-induced cell death depends on RIPK3 and MLKL.*

- In Figure 3D, the authors should explain the rationale for having an additional D mutation for some but not all phosphorylated residues. RIP1 protein levels should also be shown in order to allow comparison between the different mutants.

Response: *We are sorry that we did not have reasonable rationale for making*

additional D mutation for some residues. The mutations were not made at the same time. We made both E and D mutations at the beginning of our mutagenesis process but did not make D mutation later because the chance for having different effects between E and D mutations is low. We included RIP1 protein expression data in the revised manuscript (Fig. 3d).

- The results from Figure 4D are very interesting but it would be more appropriate to have a separate graph for each KD mutant, and each graph should contain the KD mutant alone, the KD mutant with S161E, and S161E alone.

Response: *We revised as the reviewer suggested (Fig. 4d).*

- In Figure 5D, 7B and supplemental Figure 4C-D, the results for IP RIP3 should be presented with normal contrast/illumination. In Figure 5D, it even looks like if there was an intention to hide the fact that RIP3 binds RIP1 (even at 0h) in the S161N. The resolution of Figure 5E is very bad. FADD recruitment should also be analyzed in these IPs. The quality of the IPs in supplemental Figure 4C-E is not very good.

Response: *We agree and adjusted the contrast back to normal. New experiments were performed to replace Figure 5D (Fig. 6c in revised manuscript) and Figure 7B (Fig. 6d in revised manuscript). Figure 5E was changed to Fig. 6e, and Supplemental Figure 4C-E were changed to Fig. S6d-f in the revised manuscript. We improved resolution and included FADD data as suggested.*

How do the authors explain that ROS scavenging affects recruitment of FADD and RIP3 in supplemental Figure 4D (and C??), but that the 3CS-RIP1 mutation only affects binding to RIP3 and not to FADD, and that addition of BHA still does not affect FADD binding (Figure 7C)?

Response: *In our previous version of manuscript, we showed that 3CS mutation reduced recruitment of RIP3 (Fig 7b in our old version), but we did not include data of FADD in Fig. 7b. There were no data showing that 3CS-RIP1 mutation only affects binding to RIP3 but not FADD in our previous version of manuscript. We showed in our revised manuscript, 3CS mutation also reduced recruitment of FADD (Fig. 6d in revised manuscript). It is right that BHA cannot affect FADD or RIP3 recruitment in 3CS cells because this mutation prevented cysteine oxidation (Figure 7C in old version). We think the long exposure of FADD blot shown in the figure 7C caused misunderstanding that unlike RIP3, FADD was efficiently recruited to 3CS-RIP1. We apologize for causing this confusion and replaced Figure 7C with new data (Fig. 6g, 6h in revised manuscript). We showed that BHA had no effect on KK-AT-S161E RIP1 and 3CS-S161E RIP1-mediated recruitment of RIP3 and FADD.*

Supplemental Figure 4F also seem to show that binding of RIP1 to RIP3 is not affected by BHA...

Response: *We carefully re-examined supplemental Figure 4F and believe BHA inhibited binding of RIP1 to RIP3. To better view the difference, we replaced RIP1 picture with a shorter exposure one (Fig. S6g in revised manuscript).*

- The rationale for the results presented in the supplemental part vs article is not always clear to me. Some of the work presented in the article is a confirmation of previously published studies while very important results are put in the supplemental (such as results from supplemental Figure 4-5). The results from Figure 6 are not convincing (we barely see anything), and could easily go to the supplemental part. Also, the results presented in Figure 1, 2 and 5A-B could be combined in one Figure. The results from Figure 7 on ROS/RIP1 cysteines would better fit with the first part of the story on ROS scavenging than at the end of the article.

Response: *We think the reviewer is right and re-arranged our manuscript as the reviewer suggested. The images (Figure 6 in old version) were improved (Fig. 7a, 7b, 7c in revised manuscript).*

- The authors should already refer in the introduction to the work showing that ROS promote TNF-mediated RIP1-dependent apoptosis and necroptosis by acting downstream of complex I and upstream of complex II and the necrosome (Dondelinger et al CDD 2013, Schenk et al Oncogene 2015). In p2 line 34, the authors define complex IIb and necrosome as the same complex. However, complex IIb was initially defined as a RIP1-dependent caspase-8 activating complex II (RIPK1/FADD/caspase-8) in contrast to the TRADD/FADD/caspase-8 complex IIa (Wilson et al. Nat Immunol 2009). In order to stay consistent with original definition and to avoid confusion, I would suggest to dissociate the apoptotic complex IIb from the necrosome. The authors should also check the relevance of the literature they are citing. For example, the work cited in reference to “it is also clear that RIP1 kinase activity is involved in necroptosis”¹⁷⁻²⁰ (p2 line 43) is certainly not the most logical to me.

Response: *The reviewer is right. We revised our manuscript accordingly.*

Reviewers' Comments:

Reviewer #2 (Remarks to the Author)

The authors reasonably answer to all the questions raised.

Reviewer #3 (Remarks to the Author)

The authors have adequately addressed my questions in their revised manuscript. The work is suitable for publication in NC now.

Reviewer #4 (Remarks to the Author)

The authors have substantially revised the manuscript and provided new experimental data addressing my concerns. The manuscript is, to me, now acceptable for publication at Nature Communications.

REVIEWERS' COMMENTS:

Reviewer #1 (Remarks to the Author):

The authors reasonably answer to all the questions raised.

Reviewer #2 (Remarks to the Author):

The authors have adequately addressed my questions in their revised manuscript. The work is suitable for publication in NC now.

Reviewer #3 (Remarks to the Author):

The authors have substantially revised the manuscript and provided new experimental data addressing my concerns. The manuscript is, to me, now acceptable for publication at Nature Communications.